## Registered report

psychology/cognition/behaviour

cheating, dishonesty, entitlement, feeling lucky, inequality aversion, self-confidence

**Author for correspondence:**
Andrew M. Colman
e-mail: amc@le.ac.uk

# Does competitive winning increase subsequent cheating?

Andrew M. Colman[1], Briony D. Pulford[1], Caren A. Frosch[1], Marta Mangiarulo[1] and Jeremy N. V. Miles[2]

[1]Department of Neuroscience, Psychology and Behaviour, University of Leicester, University Road, Leicester LE1 7RH, UK
[2]Keck School of Medicine, University of Southern California, 1975 Zonal Avenue, Los Angeles, CA 90033, USA

AMC, 0000-0003-0499-3479; BDP, 0000-0001-8636-9957;
CAF, 0000-0003-4153-8571; MM, 0000-0001-7995-3717;
JNVM, 0000-0002-3229-6235

In this preregistered study, we attempted to replicate and substantially extend a frequently cited experiment by Schurr and Ritov, published in 2016, suggesting that winners of pairwise competitions are more likely than others to steal money in subsequent games of chance against different opponents, possibly because of an enhanced sense of entitlement among competition winners. A replication seemed desirable because of the relevance of the effect to dishonesty in everyday life, the apparent counterintuitivity of the effect, possible problems and anomalies in the original study, and above all the fact that the researchers investigated only one potential explanation for the effect. Our results failed to replicate Schurr and Ritov's basic finding: we found no evidence to support the hypotheses that either winning or losing is associated with subsequent cheating. A second online study also failed to replicate Schurr and Ritov's basic finding. We used structural equation modelling to test four possible explanations for cheating—sense of entitlement, self-confidence, feeling lucky and inequality aversion. Only inequality aversion turned out to be significantly associated with cheating.

## 1. Introduction

In an interesting and influential article, Schurr & Ritov [1] reported the results of a series of experiments suggesting that winners of pairwise competitions are more likely than others to steal money in subsequent games of chance against different opponents. They suggested that this effect occurs only when

winning the initial competition results from superior competitive performance relative to another person (studies 1 and 2) and not from chance or luck (study 3a), nor when it reflects the achievement of a goal (study 3b). Finally, they suggested that a possible mechanism underlying the effect is an enhanced *sense of entitlement* among competition winners.

There are several reasons why it seems desirable to attempt a replication of the basic finding that competitive winning increases subsequent cheating.[1] The first is the importance of the effect in terms of societal impact, given its potential relevance to understanding the effects of winning on cheating in everyday life. The findings may be especially timely in light of current concerns about increasing academic dishonesty in the digital age [2], burgeoning problems of tax avoidance and evasion by wealthy people in developed economies [3], and more generally effects of widening inequality in wealth and income on corruption and crime [4,5].

A second reason why replication is desirable is the counterintuitivity of the reported effect. As Schurr & Ritov [1] acknowledged, we might expect the opposite effect—losers being more likely to use dishonest means to compensate themselves after being deprived of resources by winners. In fact, there is experimental evidence that losing does indeed tend to provoke dishonest behaviour [6–8], perhaps because of resulting aversive social comparison and feelings of being wronged. Contrary to the interpretation of Schurr and Ritov, Zitek *et al.* [9] suggested that it is a feeling of being wronged, rather than competitive winning, that tends to engender a sense of 'victim entitlement'. Schurr and Ritov offered no cogent explanation for the discrepancy between their findings and those of earlier researchers.

A third reason to attempt a replication is the existence of possible problems and anomalies in Schurr & Ritov's [1] studies. One problem is that most of their studies seem to have been underpowered. For example, a baseline study with only 46 participants was used to establish whether cheating occurs in a dice game against an opponent in the absence of prior competitive winning or losing. The dice game was designed to enable participants to cheat and was also used in studies 1, 2, 3a and 3b, where it was preceded by independent variable manipulations of competitive winning/losing or success/failure. The results of the baseline study were used as a 'control group' against which to compare cheating in each of these other studies. In the baseline study, the participants were university students, and half of them served merely as dyad opponents (passive recipients) whose cheating behaviour was not examined; hence, the true sample size was $n = 23$. The true participants rolled two dice and reported the total score obtained, taking the same number of Israeli shekels from a packet of 12 as payment. They were told: 'The rest of the money will go to one of the participants sitting in the lab who did not play the two-dice-under-a-cup game' [1, p. 1757]. The data revealed no evidence of cheating in the dice game, based on the fact that there was no evidence of participants on aggregate claiming scores above the chance level of 7 in the roll of two dice: 'On average participants claimed [mean $(M) = 7.13$, s.d. $= 2.46$]. Overall, participants' claims did not significantly differ from the expected 7 NIS [$t_{22} = 0.26$, $p = 0.80$]' [1, p. 1755].

A standard power analysis using G*Power [10] reveals that the achieved power in the baseline study was very low for a one-sample $t$-test: $1 - \beta$ error probability $= 0.057$. Assuming two-tailed statistical testing, to achieve a conventional power level of $1 - \beta = 0.80$, a sample size of 199 would be required to detect a small effect[2] and 34 to detect even a medium effect, based on Cohen's criteria for small ($d = 0.20$), medium ($d = 0.50$), and large ($d = 0.80$) effect sizes [11]. The study therefore appears underpowered for detecting cheating in the absence of prior competitive winning or losing. Similarly, in their study 1, used to establish the main finding that competitive winning increases cheating, the researchers reported that winners claimed significantly more than the expected amount, whereas losers' claims 'did not significantly differ from the expected 7 NIS [$t_{22} = -1.00$, $p = 0.328$]' [1, p. 1756], but a power analysis reveals, once again, that the achieved power was very low: $1 - \beta = 0.159$. With a true $n = 23$, the study was underpowered to detect a small or even a medium effect. Studies 2 (true $n = 38$), 3a (true $n = 51$) and 3b (true $n = 44$) appear to have been similarly underpowered.

The researchers also reported that winners in study 1 cheated significantly more than participants in the baseline 'control group', but participants were not assigned randomly between the baseline control group study and study 1, hence the results of their independent-samples $t$-test cannot necessarily be

---

[1]We submitted this article as a registered replication to *Proceedings of the National Acadamy of Sciences USA* (*PNAS*), the journal that published the original research of Schurr & Ritov [1]. We received an immediate desk rejection from the Managing Editor explaining that the journal would consider only a completed replication: 'As such, your PNAS submission . . . has been withdrawn from the PNAS online submission system' (email dated 8 October 2020).

[2]A sample size of 299 was mentioned in the stage 1 submission, but that was a typographical error that we correct here. Also, in the previous sentence, we did not remind readers that it was for a 'one-sample' $t$-test.

attributed to their experimental manipulation. They also reported the results of an independent samples *t*-test comparing winners and losers, suggesting that winners claimed significantly higher dice-roll scores than losers, but that result cannot on its own support a valid inference that winning *per se* increases subsequent cheating, because it could, for example, be a consequence of losing inhibiting cheating that would otherwise have occurred, or of winning increasing cheating significantly more than losing. All that can be inferred is that winners cheated more than losers; it cannot be inferred that winning and not losing led to increased cheating. None of the statistical test results appears to provide evidence to support that key hypothesis. Similar analyses were performed in studies 2, 3a and 3b, and the same criticisms apply to those studies.

Study 3b, in which participants either achieved or failed to achieve a goal before playing the dice game, seems potentially problematical for an additional, entirely different reason. The 44 participants in that study answered 20 trivia questions, and the treatment variable was manipulated as follows: 'We told participants that those who answered more than 10 questions correctly would receive a pair of earbuds. To avoid selection bias we randomly assigned participants as achievers or nonachievers and then informing half of them that they had achieved the criterion' [1, p. 1758]. It is difficult to understand how this deception could work for all participants, because it seems easy enough for participants to remember whether or not they answered more than half the questions correctly, especially if they performed much better or much worse than that. The researchers anticipated this objection and answered it as follows: 'The questions were difficult, and it was not possible for the respondents to track the number of questions they answered correctly' [1, p. 1758]. However the fact that the questions were difficult seems to compound the problem rather than solve it, because it implies that few if any participants came close to answering half the questions correctly, making it relatively easy for them to remember that they did not reach the criterion of half correct, and therefore at least some of those who were randomly assigned to the successful group may surely have been aware or suspected that they were not real achievers. No manipulation check for the success of this deception is reported in the article.

A final reason why a replication seems desirable is that Schurr & Ritov [1. p. 1754] briefly examined only a single potential explanation for the effect, namely that 'one may expect that the increased prominence of social comparison in competition will evince a sense of entitlement among winners. . . . The sense of entitlement, in turn, facilitates dishonest behavior among winners'. The researchers did not attempt to test this hypothesis fully: they provided evidence in study 4 that recalling winning a competition is associated with higher sense of entitlement compared with people who recalled achieving a goal, but they offered no evidence that a higher sense of entitlement increases subsequent cheating. In their supplemental materials, competence was investigated as a potential factor, and they seem to rule this out as an explanation.

If the effect reported by Schurr & Ritov [1] is real, and competitive winning does indeed increase subsequent cheating, then there are at least two other possible explanations for it, in addition to their *sense of entitlement* hypothesis. A second possible explanation rests on established findings that success increases people's *self-confidence* [12,13]. Taken in conjunction with the conjecture, for which there is some indirect evidence [14], that breaking a moral rule by cheating may be easier for people who are feeling self-confident, this would provide a reasonable explanation for the effect. Perhaps winners are emboldened to cheat by their increased self-confidence, and if that is the case, then competitive winning may increase self-confidence and subsequent cheating irrespective of any effect on sense of entitlement. A third possible explanation is that winning may tend to induce a psychological state of *feeling lucky*. A well-documented version of this is the hot hand fallacy [15–17], according to which winning causes some people to believe that they are on a winning streak that is likely to continue and that they will therefore probably be lucky in a subsequent game of chance, or that they are having a lucky streak that may persist for some time. If their expectations are confounded and they immediately lose, they may then cheat to reduce the resulting cognitive dissonance [18,19] between the belief that they are in a winning streak and the knowledge that they lost.

If, on the other hand, it turns out that losing (rather than winning) increases subsequent cheating, then a persuasive explanation for this, in addition to the *victim entitlement* hypothesis already mentioned, arises from experimental evidence of *inequality aversion* as a powerful and pervasive influence on decision making in experimental games [20,21], suggesting that participants who lose in initial pairwise competitions are likely to be motivated to restore equality in the subsequent dice game, and that may induce some of them to cheat.

The empirical studies reported in the sections that follow were designed to establish the replicability of the basic effect reported by Schurr & Ritov [1] and, if the replication turned out to be successful, to test the possible explanations of it outlined above, including the explanation suggested by Schurr and Ritov

themselves. We did not exclude the possibility of finding that losing (rather than winning) increases subsequent cheating [6–9].

Study 1 was designed to replicate Schurr and Ritov's study 1 as closely as possible, but with adequately increased power and an appropriate control group. The task used by these researchers to induce experiences of winning and losing was a perceptual test taken from Haran *et al.* [22]. This test is unpublished, but we obtained a copy of it from the authors. Deception is normally considered taboo in judgment and decision-making research [23–25], hence participants were randomly designated winners or losers, as in the original experiment, because we thought it necessary to replicate the original study as faithfully as possible on this point, but in a way that avoided lying to them and that also eliminated the implausibility problem with Schurr and Ritov's methodology.

The two competing hypotheses for study 1 were as follows. Hypothesis 1 (following Schurr & Ritov [1]): winners in the first, skill-based competitive task will cheat significantly more frequently in the second, chance-based competitive task than losers, and also more than control-group participants who performed the first task non-competitively. Hypothesis 2: losers in the first, skill-based competitive task will cheat significantly more frequently in the second, chance-based competitive task than winners, and also more than control-group participants who performed the first task non-competitively.

In the event, neither hypothesis 1 nor hypothesis 2 was confirmed in study 1. For study 2, we performed an online experiment, using the Gorilla Experiment Builder cloud-based research platform, in another attempt to replicate Schurr & Ritov's [1] basic effect, and we also used techniques of structural equation modelling (SEM) for this second part of study 2 to test the *sense of entitlement*, *self-confidence*, *feeling lucky*, and *inequality aversion* hypotheses. In other words, we included all the hypothesized variables that we would have included if either hypothesis 1 or hypothesis 2 had been confirmed in study 1. We present the SEM findings as a proof-of-concept investigation capable of drawing tentative conclusions, pending controlled experiments that can corroborate its findings decisively.

There were reasons to expect cheating rates to be higher in study 2 than in study 1, because the study 2 data were collected online in an unmonitored task. There is evidence that this tends to elicit more cheating than laboratory studies [26,27], presumably because it virtually eliminates residual concerns about being caught out, but this applies equally to participants in all treatment conditions, hence aggregate results can still reveal differences that provide valid tests of the hypotheses.

The hypotheses for study 2 are as follows. Hypothesis 3: cheating by competition winners in a subsequent game of chance is associated with a sense of entitlement. Hypothesis 4: cheating by competition winners in a subsequent game of chance is associated with self-confidence. Hypothesis 5: cheating by competition winners in a subsequent game of chance is associated with feeling lucky. Hypothesis 6: cheating by competition losers in a subsequent game of chance is associated with victim entitlement. Hypothesis 7: cheating by competition losers in a subsequent game of chance is associated with inequality aversion.

# 2. Material, methods and results

## 2.1. Study 1

### 2.1.1. Participants

Schurr & Ritov [1] reported a large effect size in their experiment 1, the experiment that is being replicated here. A power analysis reveals that, given $\alpha = 0.05$ and very high power $(1 - \beta = 0.95)$, to detect a large effect $(f = 0.40)$, we needed a total sample size of at least 102, and for a medium effect $(f = 0.25)$, we needed a sample size of at least 252. In the event, we recruited 259 undergraduate students on the university campus. The mean age was 19.76 years (s.d. = 2.38) and the sample was 73% female. Remuneration of up to £3.00 was paid to winners, close to the 12 Israeli shekels paid by Schurr and Ritov. The remuneration corresponds to the approximate price of a cup of coffee and a croissant, as in the original study.

### 2.1.2. Experimental design

Participants were assigned randomly to three treatment conditions: winners, losers, and control. Winners and losers were designated by first performing Haran *et al.*'s perceptual task [22] in pairs and then being assigned randomly to conditions as winners or losers. We told the participants that whether they

performed better than their opponents would determine whether they were winners or losers, and each of the winners was rewarded with a pair of earbuds. Haran *et al.*'s task is entirely unlike anything that participants are likely to have encountered before, making it very difficult for them to judge how well they had performed relative to others, and that minimized the likelihood that any of them doubted their designation as a winner or a loser (thus avoiding the methodological problem in Schurr & Ritov's [1] experiment, mentioned earlier), although those assigned to the winner and loser conditions were, in fact, bound to win or lose (see below). Control group participants performed Haran *et al.*'s perceptual task individually, without competitive pairing or any mention of competition. Participants in all three groups then played the dice-under-a-cup game used in the original experiments.

### 2.1.3. Materials

The written instructions reported in Schurr & Ritov [1] were used, apart from required changes, with instructions modified appropriately for the control group. For the dice game, a paper cup with a spyhole in its base, a pair of dice,[3] and a packet containing £3.00 in coins were provided to each participant.

### 2.1.4. Procedure

The procedure followed as closely as possible the procedure reported in Schurr & Ritov [1], apart from their unnecessary pairing of each true participant with a 'passive recipient' whose data were not used in their original study. Participants were tested in groups (median group size 18, mode 19).[4] In each testing session, participants were assigned randomly to the three treatment conditions. All participants were told at the beginning of each testing session that they would be completing two entirely different tasks during the session. They completed Haran *et al.*'s perception task [22] immediately before the dice game, but the instructions for the dice game made clear that it was a distinct and different task.

Participants in the winners and losers conditions were first told that they were each paired with another participant, and they then performed Haran *et al.*'s perception task [22]. Without deception, two additional participants (accomplices of the experimenters), trained in advance, were included in each testing session. One half of the true participants were randomly paired with a trained participant who deliberately achieved a zero score, and the other half were paired with a different trained participant who deliberately achieved a perfect score, ensuring that half the true participants were winners and half were losers. Nothing that the participants were told was untrue (each was indeed paired with another participant), although the fact that each was paired with just one of two trained participants was withheld from them. Withholding information does not violate the ethical codes of the British Psychological Society or the American Psychological Association.

It is important to point out that, from the point of view of the participants, our modification of the procedure (avoiding having half the participants not participating at all) is a distinction without a difference. Our participants did not know that each of them was paired with one of just two trained dummy participants for the dice game. This difference in procedure relates to something entirely hidden from the participants that cannot easily explain any difference between the results of Schurr & Ritov's [1] experiment and ours, but we shall return to this issue in the Discussion.

After performing the perception task, all participants in the winners and losers conditions were informed whether they were winners or losers in competition with their dyad partners. Winners were asked to come to the front of the laboratory, where they were each given a pair of earbuds as a reward. All participants were then told that they would each be paired with a different person in the same testing session (in fact, each was paired with the trained participant that they were not paired with before) and they used the paper cups with spyholes in their bases to play the dice game, almost exactly as in Schurr & Ritov [1]. Participants assigned to the control group performed the initial perceptual task without being told that they were in competition with any other person. After completing the perceptual task, they then played the dice game.

The instructions were as follows [text in square brackets was omitted for control group participants]: '[You are now paired with someone else in the room who is not the person you were paired with in the

---

[3]We checked the fairness of the dice and determined that the average score on multiple rolls of the dice did not significantly differ from the expected chance value, $t_{49} = 0.72$, $p = 0.48$.

[4]Participants chose where to sit and unfilled desks in a testing session are the explanation of non-consecutive participant numbering in the datasets. We did not delete any participants' data.

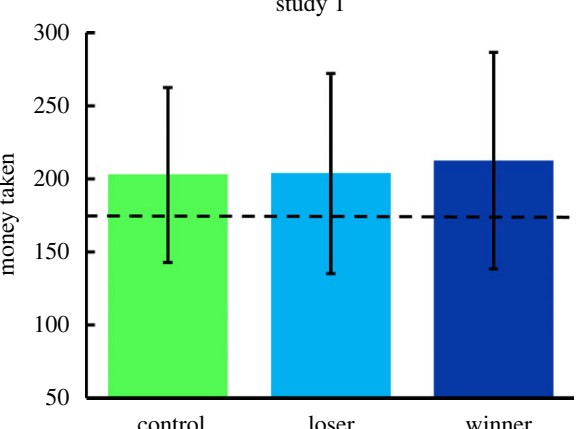

**Figure 1.** Amount of money taken by participants in the three treatment conditions of study 1. The dashed horizontal line indicates the expected value without cheating (175 pence). Error bars ± 1 s.d.

first experiment.] In this second experiment, you will perform a game of chance using two dice. You have received a cup with a hole in the bottom, two dice, and an envelope containing 3 pounds. Please turn the cup over the dice, shake it vigorously, and look down through the hole to see the result. The outcome of the shake (the numbers that came out) will be paid to you at 25p per dice spot. For example, if you rolled two 1s (a score of $2 \times 25p$) you will get 50p; if you rolled two 6s (a score of $12 \times 25p$) you will get 3 pounds. [The rest of the money in the envelope will go to the other person in the lab you are paired with.] Once you have rolled and taken your money you may leave the room as the experiments are over. Please leave the envelope and cup and dice on your desk. Thank you.' We inferred the participants' claimed dice-roll scores from the amounts of money that they removed from their envelopes.[5]

### 2.1.5. Results

No data were excluded from analysis. The findings failed to support either of our hypotheses (hypotheses 1 and 2). The outcome variable (the amount of money claimed by each true participant in the dice game) was compared across treatment conditions using one-way between-subjects ANOVA. The amount of money taken by the participants in the three conditions did not differ significantly (control $M = 202.78$, s.d. $= 59.76$; loser $M = 203.66$, s.d. $= 68.40$; winner $M = 212.53$, s.d. $= 74.00$); $F_{2,256} = 0.546$, $p = 0.580$, partial $\eta^2 = 0.004$; figure 1.

In each treatment condition, one-sample $t$-tests were also used to test the significance of reported dice-roll scores relative to chance expectation of 175 pence (a mean roll of 7 multiplied by 25 pence). All three groups claimed significantly more than the expected chance amount; control group: $t_{87} = 4.361$, $p < 0.001$; loser: $t_{85} = 3.886$, $p < 0.001$; winner: $t_{84} = 4.676$, $p < 0.001$. A one-tailed one-sample $t$-test collapsing across all four groups showed significant overall cheating in study 1; $M = 206.27$, s.d. $= 67.44$, $t_{258} = 7.463$, $p < 0.001$, effect size $d = 0.46$ (small to medium).

## 2.2. Study 2

After testing participants in study 1, we became aware of a potential confound in that study. The control group participants in study 1 performed the preceding visual task, which makes the control group more comparable with the winner and loser groups and improves the design. However, in our study 1 control

---

[5]According to Schurr & Ritov [1], 'First, we examined the amount of self-reported winnings participants claim in the dice-under-a-cup task' [1, p. 1754], and 'The participant places the cup over the dice, shakes it, and receives money according to his or her reporting of the outcome of the roll' [1, pp. 1754–1755]. However, an anonymous reviewer with detailed knowledge of the original study has revealed that the participants did not actually report the outcomes of the rolls other than by removing money from their envelopes: 'In the original study participants playing the dice under cup game paid themselves from the envelop [sic] and left the lab without reporting their earnings to the experimenter. The experimenter then counted how much money was left in each envelop [sic] and distributed the envelops [sic]' (reviewer's comment received on 26 March, 2021). We decided to follow the same procedure, because participants may be more tempted to cheat when they can do so without lying and can justify it as carelessness rather than dishonesty.

group, participants did not have a co-player in the dice task but did have an unspecified co-player in the experimental groups. As Schurr and Ritov discussed in relation to their research: 'Notably, in contrast to most experiments on dishonest behaviour, participants who cheated stole money from their counterparts rather than from the experimenter' [1, p. 1755]. In Schurr and Ritov's study, the control group performed the baseline dice-under-a-cup task with an unspecified co-player who lost money if the participant cheated, to mirror identically the situation in the experimental winner and loser conditions. Only then can the influence of winning and losing and feelings of entitlement be clearly identified. Thus, in addition to our *unpaired control group* (control 1) we decided to add a second *paired control group* (control 2) to study 2 with an unspecified co-player who lost money whenever a participant cheated. We also retained the unpaired control condition, without a co-player (as in study 1) for comparison, to determine whether this factor has any effect on cheating.

### 2.2.1. Participants

A power analysis revealed that, given $\alpha = 0.05$ and very high power $(1 - \beta = 0.95)$, to detect a large effect $(f = 0.40)$, we needed a total sample size of at least 100, and for a medium effect $(f = 0.25)$, we needed a sample size of at least 200. We recruited 275 university students, for comparability with study 1, and assigned them randomly to the four treatment conditions: unpaired control 1, paired control 2, winners, losers. The mean age was 20.65 years (s.d. = 3.01), and the sample was 61% female. Remuneration was paid to winners according to the random lottery incentive system, a method that avoids problems associated with other incentive schemes [28] and has been shown to elicit behaviour in line with true preferences [29,30]. Participants were told that 1 in 20 of them, selected at random, would be paid £5 in Amazon gift vouchers for every coin-flip showing heads that they reported, resulting in payments from zero to a possible maximum of £50 for 10 heads claimed. Fourteen winners received payments ranging from £10 to £40 (M = £27.14).

### 2.2.2. Materials

Because we did not expect all potential online participants to have suitable materials for playing the dice game used in study 1, the game of chance in this study involved tossing a coin 10 times, with £5 awarded for every time heads came up. Although the study was conducted online, using the Gorilla Experiment Builder platform, participants performed the coin flip task with real coins of their own, because an online coin-tosser might have raised suspicions of covert surveillance and of a possibility of being caught out if they cheated. The range of possible scores was 0–10, comparable to the range in the dice game in study 1, in which rolling two dice yielded a range of 2–12. Participants were asked to report the total number of heads that they achieved in 10 coin flips, so that cheaters would have, in effect, to tell only one lie, as in the dice game.

The first study did not reveal significantly more cheating in any group relative to another. Thus, we were unable to exclude the hypotheses that either winning or losing is associated with subsequent cheating. Nevertheless, we did find significant levels of cheating; hence, in an attempt to explain what drives cheating, we measured all hypothesized psychometric factors and included them in the analysis of results. To test the sense of entitlement, self-confidence, inequality aversion and feeling lucky hypotheses, all participants responded to four psychometric scales. Sense of entitlement was measured with the Psychological Entitlement scale [31], self-confidence with the internal confidence items of Self-Confidence scale [32], personal luckiness with the Belief in Luck and Luckiness scale [33] and inequality aversion with the inequality aversion items from slider version of the Social Value Orientation scale [34].

### 2.2.3. Procedure

Participants, apart from those in the two control groups, were first told that they were each paired with another participant in the testing session and would perform Haran *et al.*'s perception task [22]. In each testing session, participants were assigned randomly to the four treatment conditions (winners, losers, paired control, unpaired control) by the Gorilla software. Each of the winners was informed that they had been rewarded with a pair of earbuds that they should collect after the testing session. Participants in the control groups performed the perceptual task without being told that they were paired with another participant and without any suggestion of competition. Participants in the

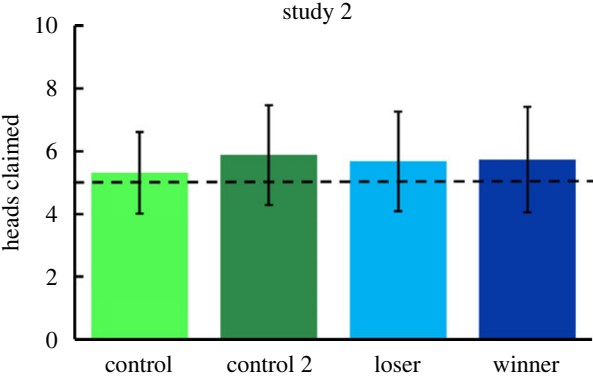

**Figure 2.** Number of heads claimed by participants in the four treatment conditions of study 2. Control 1 is unpaired, control 2 is paired with an unspecified co-player as in the loser and winner conditions. The dashed horizontal line indicates the expected value without cheating (5). Error bars ± 1 s.d.

unpaired control group did not have a co-player in the coin-flip task whereas those in the paired control group did have a co-player who would receive remaining money not claimed by the participant.

After completing the perception task and (apart from the control groups) being designated as winners and losers, all participants were then told they were each being re-paired with a different person to play the coin-flip task and that the rest of the money, after they took their winnings, would go to the other person that they were paired with (except in the unpaired control condition). Half the participants completed the four psychometric scales described above immediately and half completed the scales after the coin-flip task. This was to enable us to check whether the association between winning/losing and sense of entitlement and the other psychological variables fades rapidly and whether the association between these variables on cheating behaviour fades rapidly. After all testing sessions were complete, 1 in every 20 participants were randomly selected and paid in Amazon gift vouchers according to how many heads they reported having tossed.

For study 2, acknowledging that more than one of the hypothesized explanations for the effect may have some validity, several hypotheses were tested with SEM. We designed an SEM model in which group membership is predictive of two latent variables, both associated with number of heads claimed and hence cheating, that we labelled *pride* and *shame*. Pride is indicated by the measured variables *confidence*, *luckiness* and *entitlement*, and shame by the measured variables *entitlement* and *inequality aversion*. These variables were hypothesized to mediate the relationship between the experimental intervention and cheating (the outcome). There were no missing data in our analysis.

### 2.2.4. Results

Data from all 275 participants were included in the analyses. The outcome variable (the number of heads claimed in the coin-flip task) was compared across treatment conditions and the order of tasks (coin-flip first or questionnaires first) using a two-way between-subjects ANOVA. The difference across the four conditions did not reach statistical significance (control 1, $M = 5.31$, s.d. $= 1.30$; control 2, $M = 5.88$, s.d. $= 1.59$; loser, $M = 5.68$, s.d. $= 1.59$; winner, $M = 5.73$, s.d. $= 1.68$), $F_{3,267} = 1.743$, $p = 0.159$, partial $\eta^2 = 0.019$; figure 2. The main effect of the order of the tasks was not significant, $F_{1,267} = 1.497$, $p = 0.222$, partial $\eta^2 = 0.006$, and the interaction between condition and task order was not significant, $F_{3,267} = 1.069$, $p = 0.363$, partial $\eta^2 = 0.012$.

In each treatment condition, one-sample *t*-tests were also used to test the significance of reported coin-flip scores relative to chance expectation of 5 (the mid-point on the 0–10 range). A one-tailed *t*-test is reported because we expected significantly higher scores than 5, based on the results of study 1. All four groups claimed significantly more than the expected chance amount: control, mean difference $= 0.31$, $t_{67} = 1.965$, $p = 0.027$; control 2, mean difference $= 0.88$, $t_{68} = 4.630$, $p < 0.001$; loser, mean difference $= 0.68$, $t_{67} = 3.514$, $p < 0.001$; winner, mean difference $= 0.73$, $t_{69} = 3.637$, $p < 0.001$. A one-tailed one-sample *t*-test collapsing across all four groups showed significant overall cheating in study 2, $M = 5.65$, s.d. $= 1.55$, $t_{274} = 6.963$, $p < 0.001$, effect size $d = 0.42$ (small to medium).

Our results failed to support hypotheses 3, 4, 5 or 6 but they provided weak support for hypothesis 7. The number of heads claimed was not significantly correlated with psychological entitlement ($r = 0.09$,

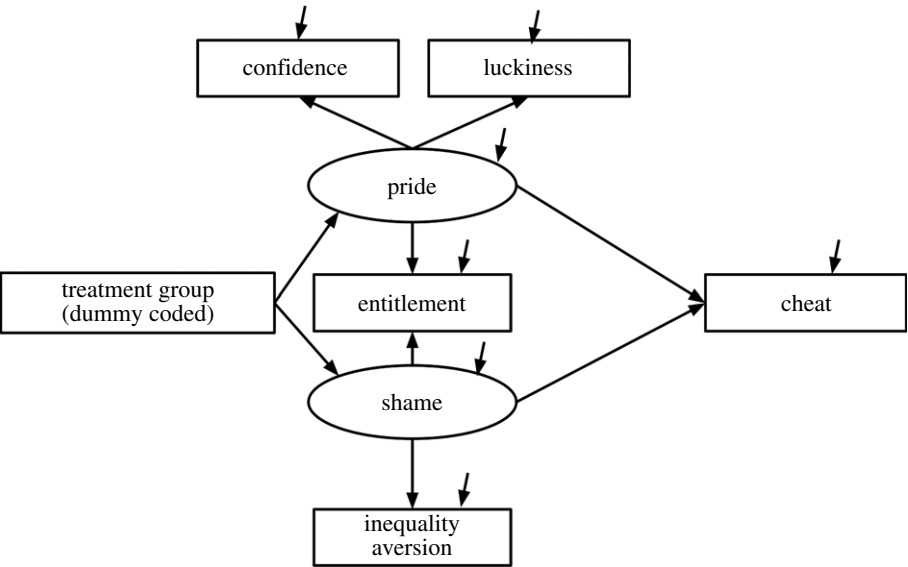

**Figure 3.** Path diagram of the structural equation model.

**Table 1.** Descriptive statistics (means ($M$) and standard deviations (s.d.)) for the personality variables split by condition.

|  | control 1 | | control 2 | | loser | | winner | |
|---|---|---|---|---|---|---|---|---|
|  | $M$ | s.d. | $M$ | s.d. | $M$ | s.d. | $M$ | s.d. |
| personal luckiness | 3.08 | (0.54) | 2.98 | (0.77) | 3.21 | (0.77) | 3.18 | (0.70) |
| entitlement | 3.05 | (1.16) | 3.19 | (1.01) | 3.06 | (1.08) | 2.88 | (1.12) |
| internal confidence | 4.57 | (1.00) | 4.51 | (1.08) | 4.77 | (1.03) | 4.89 | (1.11) |
| inequality aversion | 0.12 | (0.13) | 0.16 | (0.15) | 0.16 | (0.16) | 0.15 | (0.15) |

$p = 0.13$), internal confidence ($r = -0.03$, $p = 0.61$), or personal luckiness ($r = 0.04$, $p = 0.51$). However, the number of heads claimed did show a weak relationship with inequality aversion, $r_{275} = 0.14$, $p = 0.02$, when all four conditions were combined; on this measure, higher scores indicate lower inequality aversion, hence participants who were least inequality averse (those with the highest scores) cheated the most.

We fitted the SEM [35] using maximum-likelihood estimation, and found that it fit the data well ($\chi_{12}^2 = 11.1$, $p = 0.520$, CFI = 1.00, RMSEA = 0.00, SRMR 0.03). The path diagram is shown in figure 3. However, the only path that reached statistical significance was the path from *shame* to *cheat* (standardized estimate = 0.30, $p = 0.032$), hence this modelling approach did not provide any additional information. The initial model contained one Heywood case (negative variance), which was not statistically significant. Constraining this value to zero did not change the model parameters in any notable way.

### 2.2.5. Unregistered *post hoc* results

The reliabilities of the scales were all high. Cronbach's $\alpha$ values were: psychological entitlement (0.86), internal confidence (0.90), personal luckiness (0.83) and inequality aversion (0.89). There were no significant gender differences between the number of heads claimed by men ($M = 5.48$, s.d. = 1.64) and women ($M = 5.75$, s.d. = 1.50), $t_{269} = 1.411$, $p = 0.16$.

There were no significant differences between the four groups in their levels of psychological entitlement, $F_{3,271} = 0.966$, $p = 0.409$, $\eta^2 = 0.011$, internal confidence ($F_{3,271} = 1.895$, $p = 0.131$, $\eta^2 = 0.021$), personal luckiness ($F_{3,271} = 1.505$, $p = 0.214$, $\eta^2 = 0.016$), or inequality aversion, $F_{3,271} = 1.559$, $p = 0.200$, $\eta^2 = 0.017$; table 1.

# 3. Discussion

Schurr & Ritov's [1] experiments were severely underpowered and vitiated by other design and methodological problems. In particular, their basic finding that competitive winning is associated with subsequent cheating was based on a study in which participants were not assigned randomly to experimental and control treatment conditions. Our study 1 replicated Schurr and Ritov's study as closely as possible with adequate power and random assignment to experimental and control conditions. We observed significant levels of cheating in both experimental and control conditions but failed to replicate Schurr and Ritov's basic finding of higher cheating by winners, although the experimental manipulation of winning or losing in both of our experiments was identical to Schurr and Ritov's. We also found no evidence for any significant effect of competitive losing on cheating in the subsequent game of chance.

In study 2, we tested the hypotheses that competitive winning or losing is associated with subsequent cheating in an even larger experiment, conducted online, with participants assigned randomly to winning, losing, paired control, and unpaired control treatment conditions. Once again, we observed significant levels of cheating in all treatment conditions but found no evidence to support the hypotheses that either winning or losing is associated with subsequent cheating. There was no significant difference in cheating between our paired and unpaired control conditions—whether cheating was associated with money being taken from another participant or from the experimenter.

This study also included an investigation, using SEM, to test the hypotheses that winning is associated with a latent variable that we labelled 'pride', indicated by self-confidence, a feeling of luckiness, and a sense of entitlement, and that pride is associated with subsequent cheating, or that losing is associated with a latent variable of 'shame', indicated by a sense of entitlement and inequality aversion, and that shame is associated with subsequent cheating. We measured all the indicator variables with psychometric scales that showed high reliability in our study, and the only significant association that emerged was between inequality aversion and cheating. This suggests that participants who were least inequality-averse were most likely to cheat in the coin-flip game, whether they had won or lost the previous competitive perceptual task. The association of inequality aversion with cheating was not strong, but it is worth investigating experimentally. It may reflect a more general sense of fairness among participants who are inequality averse. If those who value fairness strongly tend to be inequality averse and also construe cheating as a form of unfairness, the association would be explained, but that explanation requires further experimental evidence.

One key question that needs to be addressed is why the results of both of our studies failed to replicate Schurr & Ritov's [1] basic finding that competitive winning is associated with subsequent cheating in a game of chance. One possibility is that Schurr and Ritov's finding, based as it was on a severely underpowered study without proper random assignment to experimental and control groups, cannot validly be inferred from their results. A second possibility relates to their unusual methodology, in which half the participants in every testing session were randomly assigned as passive participants, whose only role was to receive the money that the other half—the active participants—left behind after taking what they claimed was owing to them after the dice game. In our studies, the participants were told that the money that they left behind would go to 'the other person you are paired with', with the implication that it was one of the other participants, and in that sense, from the participants' point of view, it was similar to what Schurr and Ritov's participants believed. However, the active participants in Schurr and Ritov's experiment believed that the money they left would go to others who had done absolutely nothing in the experiment, whereas the participants in our replications could have believed that the money would go to others who were fully participating. All of Schurr and Ritov's participants who rolled the dice were told that 'The rest of the money will go to one of the participants sitting in the lab who did not play the two-dice-under-a-cup game' [1, p. 1757]. This might possibly explain the failure of our experiments to replicate Schurr and Ritov's basic finding if their winners, in contrast to ours, believed that the recipients of money left behind were more deserving of being cheated, but that would suggest that the basic finding applies only in the artificial context of their experimental setup or in very limited and unusual circumstances. In everyday life, people who cheat rarely, if ever, know that their victims have done nothing to earn the money out of which they are being cheated. A third possibility is that the discrepancy between Schurr and Ritov's findings and ours arises from a cross-cultural difference between students in Israel and the UK; but we are unaware of any evidence that might support that interpretation, it is very unlikely given that Israel and the UK are both WEIRD (western, educated, industrialized, rich, democratic) cultures [36], and if correct it would

severely limit the generality of Schurr and Ritov's basic finding (and also, by symmetry, our own basic finding).

Given the published evidence that more cheating tends to occur in online than laboratory studies [26,27], it is worth noting that we found no such difference. In study 1, incentives were lower than in study 2 and participants cheated, on average, by 31p out of a possible maximum of £3.00 (10.3%), while in study 2 they cheated by £3.25 out of a maximum of £50.00 (6.5%). The incentives were much greater in study 2, therefore participants' cheating translated into greater monetary terms in study 2. In our laboratory-based study 1, using Cohen's [11] index of effect size, the overall effect size of cheating was $d = 0.46$ and in our online study 2 it was $d = 0.42$. The finding of such a negligible difference can perhaps be explained by the fact that the dice-under-a-cup game that we used in the laboratory in study 1 provides an opportunity for cheating that seems almost entirely 'safe', in the sense that it would be impossible to detect a particular instance of cheating. If that interpretation is right, then our online experiment, in which the corresponding task was a coin-flip task, may not have provided a significantly greater sense of security, and participants may have felt equally disinhibited from cheating in both experiments. Thus the general cheating that we found across all conditions would be in line with recent evidence that cheating tends to occur particularly when it is unobservable by the experimenters [37].

Our studies have not provided much enlightenment as to what leads some people to cheat. In both studies, cheating occurred at a low but significant level in all treatment conditions, and the only psychometric variable that correlated significantly with cheating was inequality aversion. Our SEM revealed only one path that reached statistical significance, from *shame* to *number of heads* claimed and hence cheating. One of the indicators of shame was inequality aversion. Further research is clearly required to determine whether inequality aversion is indeed causally related to cheating and if so why. One possibility is that inequality aversion is associated with a more general concern for fairness and that people who value fairness are less likely to cheat because they perceive cheating as a form of unfairness, but without further evidence this interpretation remains speculative.

The aim of study 2 was to discover variables, possibly but not necessarily including competitive winning or losing, that might explain cheating in a subsequent game of chance. The SEM should reveal whether, and if so how, winning or losing is implicated. We hypothesized that sense of entitlement, self-confidence, personal luckiness and inequality aversion might help to explain cheating. For example, if Schurr & Ritov [1] were right, then winning should be associated with sense of entitlement and sense of entitlement should be associated with cheating. Sense of entitlement is interpreted by the authors of the scale that we used to measure it [31] as a personality trait, and we should perhaps expect a personality trait to be largely unaffected by an experimental manipulation such as winning or losing. However, the SEM does not require winning or losing to play any part in the potential relationship of any of the other variables to cheating. For example, we might have found that trait sense of entitlement is associated with cheating, irrespective of any association with winning or losing, just as we did, in fact, find that inequality aversion is associated with cheating without any significant association with winning or losing.

# 4. Conclusion

Winners of pairwise skill-based competitions do not cheat in subsequent games of chance against different opponents by stealing significantly more money than losers or control-group participants steal. Furthermore, despite evidence that losing tends to lead to dishonest behaviour in many circumstances [6–8], losers do not cheat more than winners or control-group participants cheat. These conclusions emerge from a large laboratory experiment and an even larger online experiment reported in this article, and they apply at least to the particular manner in which cheating was enabled and measured in these experiments and in the experiment by Schurr & Ritov [1] on which they were closely modelled.

Within this experimental paradigm, winning does not significantly increase winners' sense of entitlement, as hypothesized by Schurr & Ritov [1], when sense of entitlement is measured by the Psychological Entitlement scale [31]. Winning does not significantly increase winners' self-confidence as measured by the Self-Confidence scale [32] or their personal luckiness as measured by the Belief in Luck and Luckiness scale [33]. None of these variables is significantly associated with cheating. Losing does not significantly increase losers' sense of entitlement or inequality aversion, as measured by the slider version of the Social Value Orientation scale [34], but there does turn out to be a significant association between inequality aversion and cheating. Low inequality aversion is associated

with cheating, perhaps because people who value fairness highly tend to be inequality averse and thus avoid cheating.

Ethics. Ethical approval for both studies was obtained through the Psychology Ethics Committee in the Department of Neuroscience, Psychology and Behaviour of the University of Leicester. Our procedures were all fully compliant with the British Psychological Society's Code of Ethics and Conduct. Ethics Reference: 30007-mm888-ls:neuroscience, psychology & behaviour.

Data accessibility. Data, experimental materials and a laboratory log are freely available in the OSF repository: https://osf.io/672ad. SPSS and R (Lavaan) SEM data analysis syntax files are deposited in the same OSF repository. Experimental materials for study 2 are available on Gorilla Open Materials repository at the following link: https://app.gorilla.sc/openmaterials/398595. The approved stage 1 protocol is available at https://osf.io/kf7hr.

Authors' contributions. A.M.C.: conceptualization, formal analysis, funding acquisition, investigation, methodology, writing—original draft, writing—review and editing; B.D.P.: conceptualization, data curation, formal analysis, funding acquisition, investigation, methodology, project administration, resources, supervision, validation, visualization, writing—original draft, writing—review and editing; C.A.F.: funding acquisition, writing—original draft, writing—review and editing; M.M.: data curation, investigation, project administration, software, validation, writing—original draft, writing—review and editing; J.M.: formal analysis, software, writing—review and editing.

All authors gave final approval for publication and agreed to be held accountable for the work performed therein.

Conflict of interest declaration. The authors declare no competing interests.

Funding. The research reported in this article was supported by a grant from the Leicester Judgment and Decision Making Endowment Fund (grant no. M56TH33) to the first three authors.

Acknowledgements. We would like to thank Richard Kirkden for his help with the programming of the Gorilla software for study 2 and Trisha Spencer for help with data collection in study 1.

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
