## [Peer Review File · Royal Society Open Science]

Review History

RSOS-202197.R0 (Original submission)

Review form: Reviewer 1 (Shira Elqayam)

Do you have any ethical concerns with this paper?

No

Have you any concerns about statistical analyses in this paper?

Yes

Recommendation?

Major revision

Comments to the Author(s)

Review of RSOS-202197, Colman, Pulford, Frosch, & Mangiarulo: Does Competitive Winning Increase Subsequent Cheating?

This is a pre-registered study of two experiments, of which the first one is a replication study of Schurr and Ritov (2016), and the second one is a follow-up study aiming at an explanation of

whatever effects might be found in the first experiment. I enjoyed reading the manuscript a lot, and I think it has much to recommend it. Nevertheless, I would recommend some changes before it is ready for publication in RSOC. My main concern is with the second experiment, although I have some minor comments for the first experiment as well. Let me take things in order.

The literature review, rationale for the study, and alternative hypotheses, are all presented in a crystal-clear, engaging manner. The authors make a compelling case for the significance of the replication both for the academic community and more generally for society and non-academic beneficiaries. The combination, in the original study, of counterintuitive findings with lack of power is a well-established recipe for non-replicability. Add to this the societal and theoretical significance of these findings, and you have a very worthy pre-registered replication study, likely to be widely cited.

Study 1 is likewise well-designed. The authors make a strong argument about the flaws in the original S&R study, and propose a sensible way forward for amending them. I only have a couple of very minor comments here. The first one is to query the method of obtaining exactly 252 participants, given attrition. How will the authors treat attrition and how will dropout participants be replaced and when? My second comment is about the exchange rate between NIS and GBP – this seems to be calculated by the current (2020) rate, but the exchange rate changed dramatically post the 2016 Brexit vote, and S&R's study might well have been conducted prior to this. With the small sums involved this might not make a huge difference, but it's worth checking and maybe commenting on.

The main issues I see are with Study 2. The authors rightly criticise S&R for not providing evidence to support a causal mechanism for their explanation of the findings. However, their own proposed method also falls short of a fully causal explanation. Their hypotheses are stated clearly, and only aim at establishing association rather than causal mechanisms, but given the criticism of the previous work this is a bit disappointing. Admittedly, the proposed SEM analysis somewhat compensates for this, since SEM can establish causality, but the authors are right to be wary of claiming causality without experimental manipulation. Perhaps the best way to treat Study 2 is as a proof-of-concept study, which will need to be followed up with experimental design. Another tweak I would recommend is running SEM-specific power analysis – a sample size of 252 seems a bit on the short side for SEM, especially given the dichotomous IV and the potential need for multi-group analysis.

My other concern is the use of trait scales in Study 2 to measure changes following an experimental manipulation. The latter is likely to have effects on states, less likely to affect traits, which are more stable. I have no quarrel with the authors' decision not to measure these scales pre-manipulation (reactivity and carryover are too risky here), but why would they expect changes in the first place? I would like to see a rationale for using trait scales for measuring effects of experimental manipulation.

In conclusion, this is a worthy replication proposal, and the replication study is well designed. It reads well, and I have no doubt it will be of significant interest to the academic community, as well as non-academic beneficiaries of research. I do have some concerns about the follow-up study and I think it needs more work before it is ready. Nevertheless, my firm recommendation is to give the authors the opportunity to revise this second study, as the plan in its entirety is of value to the academic and non-academic community.

Signed: Shira Elqayam

Review form: Reviewer 2

Do you have any ethical concerns with this paper?

No

Have you any concerns about statistical analyses in this paper?

No

Recommendation?

Major revision

Comments to the Author(s)

The paper "Does Competitive Winning Increase Subsequent Cheating?" aims at replicating an experiment conducted by Schurr & Ritov (2016) and substantially extending and testing other plausible psychological mechanisms underlying the effect.

Before commenting on the experimental design of the replication and the subsequent experiment, it is important to put things in the right context. Schurr & Ritov (2016) examine how social comparisons winners and losers make affects their sense of entitlement and subsequent asocial behaviors. In contrast to previous research, Schurr & Ritov examine what happens after the competition ends, competitors adapt to their status as winners or losers, and they go on to interact with new partners.

Schurr & Ritov claim that a key factor influencing winners' and losers' post-competition dishonesty is the perceived relevance of the competition to the individual's self-image. To the extent that winning is perceived to be determined by the individual's ability, effort, or rights, the winner, more than the loser, may feel entitled to behave in self-serving ways. In that case, winning (as opposed to losing) is likely to engender asocial behaviors such as avoiding the opportunity to help others, or dishonestly taking more than one is entitled to. However, if winning is not fairness-based, such as when it is randomly determined, the loser, rather than the winner, may act to "correct" the unfair outcome and cheat to compensate for the perceived unfair loss.

The authors propose to examine two psychological mechanisms - self-confidence and feeling lucky. Self-confidence is a reasonable mechanism that merits examination. As per feeling-lucky, there was no indication for this mechanism. If anything, Study 3a in Schurr & Ritov demonstrates that feeling lucky enhances egalitarianism among winners and does not affect losers.

The text includes several inaccuracies and wrong interpretations:

- (1) Competing hypothesis - The authors propose a competing hypothesis that losing rather than winning increases subsequent cheating. To back their claim the authors refer to Zitek's study (see page 2 lines 5-9). Note however that Zitek and her colleagues examined situations where the competition was stopped in the middle in an unfair way (Experiment 3). In this case, as noted above, the loser rather than the winner may act to level the field. Such a behavior indicates that the competitors did not adapt to their status.
- (2) It is also inaccurate to frame Schurr & Ritov (2016) Study 3b as reflecting "the achievement of a purely personal goal". The idea behind this study is to attenuate social comparisons by focusing participants on meeting a fixed threshold.
- (3) The authors claim that in the trivia study participants could infer their actual performance in the study. It is unlikely the case, each question had 4 possible answers and participants did not receive a feedback after answering the questions. The questions were tricky in the sense that two answers seemed plausible. Importantly, to the extent to competition prompts relative judgment, the study was not used to test competition, but rather to test what happens when relative judgment attenuates and participants meet a fixed threshold.
- (4) As per the anomalies and problems you note (page 3 line 12), increasing sample size is always adequate. Importantly, back in 2015, when the study was conducted this was the common and adequate sample size.

The replication study:

The proposed 'replication study' does not actually replicate the original study, due to two major flaws:

(1) Schurr & Ritov (2016) let half the participants play the dice under cup game. In the proposed replication study all the participants play the dice game. This is a deviation from the original procedure and its consequences cannot be predicted.

(2) Pairing participants with collaborators who purposefully achieve 0 or perfect score is bound to raise suspicions among the participants.

Importantly, when the original study was conducted, the experimenter flipped a coin and assigned all participants on the right or left side of the laboratory the role of dictators or recipients respectively.

The original payment was set to 12 shekels because this was the price of a cup coffee and a croissant. Do 3 pounds have the same buying power?

Finally, we are in the middle of a COVID-19 crisis. People are required to wear facial masks and practice social distancing. These requirements are likely to affect dishonesty.

The online study:

While there is indeed place to explore different processes that may account for the effect, there are three issues the authors should pay attention to:

Cheating in online studies is problematic. It is difficult to achieve liable and replicable dishonesty in online studies that were conducted on online panels. Participants do not believe that they are playing against a real person, nor do they believe that the experimenter respects and maintains complete privacy.

In addition, paying participants with vouchers increases the chances that losers will try to compensate for their lost income. Therefore, it is important to give participants a present, without telling its monetary value.

Finally, pairing participants with collaborators who purposefully achieve 0 or perfect score raises suspicions among participants.

Decision letter (RSOS-202197.R0)

Dear Dr Mangiarulo,

The Editors assigned to your Stage 1 Replication submission ("Does Competitive Winning Increase Subsequent Cheating?") have now received comments from reviewers. We would like you to revise your paper in accordance with the referee and editors suggestions which can be found below (not including confidential reports to the Editor). Please note this decision does not guarantee eventual acceptance.

Please submit a copy of your revised paper within three weeks (i.e. by the 10-Feb-2021). If deemed necessary by the Editors, your manuscript will be sent back to one or more of the original reviewers for assessment.

To revise your manuscript, log into <http://mc.manuscriptcentral.com/rsos> and enter your Author Centre, where you will find your manuscript title listed under "Manuscripts with Decisions." Under "Actions," click on "Create a Revision." Your manuscript number has been

appended to denote a revision. Revise your manuscript and upload a new version through your Author Centre.

When submitting your revised manuscript, you must respond to the comments made by the referees and upload a file "Response to Referees" in the "File Upload" step. Please use this to document how you have responded to the comments, and the adjustments you have made. In order to expedite the processing of the revised manuscript, please be as specific as possible in your response.

Once again, thank you for submitting your manuscript to Royal Society Open Science and I look forward to receiving your revision. If you have any questions at all, please do not hesitate to get in touch. Full author guidelines may be found at <https://royalsocietypublishing.org/rsos/replication-studies#AuthorsGuidance>.

Kind regards,
Professor Chris Chambers
Royal Society Open Science
openscience@royalsociety.org

on behalf of Professor Chris Chambers (Registered Reports Editor, Royal Society Open Science)
openscience@royalsociety.org

Associate Editor Comments to Author (Professor Chris Chambers):

Associate Editor: 1

Comments to the Author:

Two expert reviewers have now assessed the manuscript, and both raise major issues about the methodology that will need to be addressed in revision (with Reviewer 2 being especially critical). First, as a point of editorial guidance, the authors should note that the Replication format is expressly intended for close (as possible) replications of previous studies, not extensions. Therefore, if the authors want to continue down the Replications track then Study 2 should be dropped from the proposal, and the methods of Study 1 must be aligned as closely as possible to the original methods (retaining only deviations that are unavoidable). Where there are potential flaws in the original target study, the authors should still conduct a close replication (regardless of flaws) but could then propose a tight follow-up replication study that addresses them (and is therefore slightly less close to the original study). The authors may therefore consider dropping Study 2 and splitting Study 1 into two studies -- one that is as close as possible to the target study (addressing the reviewers' concerns in this area), and another with minor modifications.

I appreciate that the authors may not prefer this route. If so, and they wish to preserve the current sequence of studies, then the article should be transferred from the Replications article type to the Registered Reports article type, which provides more latitude for deviations from the original replication study while also requiring more strict criteria to be met. If the authors prefer this option then please contact the journal office to discuss next steps.

Comments to Author:

Reviewer: 1

Comments to the Author(s)

Review of RSOS-202197, Colman, Pulford, Frosch, & Mangiarulo: Does Competitive Winning Increase Subsequent Cheating?

This is a pre-registered study of two experiments, of which the first one is a replication study of Schurr and Ritov (2016), and the second one is a follow-up study aiming at an explanation of

whatever effects might be found in the first experiment. I enjoyed reading the manuscript a lot, and I think it has much to recommend it. Nevertheless, I would recommend some changes before it is ready for publication in RSOC. My main concern is with the second experiment, although I have some minor comments for the first experiment as well. Let me take things in order.

The literature review, rationale for the study, and alternative hypotheses, are all presented in a crystal-clear, engaging manner. The authors make a compelling case for the significance of the replication both for the academic community and more generally for society and non-academic beneficiaries. The combination, in the original study, of counterintuitive findings with lack of power is a well-established recipe for non-replicability. Add to this the societal and theoretical significance of these findings, and you have a very worthy pre-registered replication study, likely to be widely cited.

Study 1 is likewise well-designed. The authors make a strong argument about the flaws in the original S&R study, and propose a sensible way forward for amending them. I only have a couple of very minor comments here. The first one is to query the method of obtaining exactly 252 participants, given attrition. How will the authors treat attrition and how will dropout participants be replaced and when? My second comment is about the exchange rate between NIS and GBP – this seems to be calculated by the current (2020) rate, but the exchange rate changed dramatically post the 2016 Brexit vote, and S&R's study might well have been conducted prior to this. With the small sums involved this might not make a huge difference, but it's worth checking and maybe commenting on.

The main issues I see are with Study 2. The authors rightly criticise S&R for not providing evidence to support a causal mechanism for their explanation of the findings. However, their own proposed method also falls short of a fully causal explanation. Their hypotheses are stated clearly, and only aim at establishing association rather than causal mechanisms, but given the criticism of the previous work this is a bit disappointing. Admittedly, the proposed SEM analysis somewhat compensates for this, since SEM can establish causality, but the authors are right to be wary of claiming causality without experimental manipulation. Perhaps the best way to treat Study 2 is as a proof-of-concept study, which will need to be followed up with experimental design. Another tweak I would recommend is running SEM-specific power analysis – a sample size of 252 seems a bit on the short side for SEM, especially given the dichotomous IV and the potential need for multi-group analysis.

My other concern is the use of trait scales in Study 2 to measure changes following an experimental manipulation. The latter is likely to have effects on states, less likely to affect traits, which are more stable. I have no quarrel with the authors' decision not to measure these scales pre-manipulation (reactivity and carryover are too risky here), but why would they expect changes in the first place? I would like to see a rationale for using trait scales for measuring effects of experimental manipulation.

In conclusion, this is a worthy replication proposal, and the replication study is well designed. It reads well, and I have no doubt it will be of significant interest to the academic community, as well as non-academic beneficiaries of research. I do have some concerns about the follow-up study and I think it needs more work before it is ready. Nevertheless, my firm recommendation is to give the authors the opportunity to revise this second study, as the plan in its entirety is of value to the academic and non-academic community.

Signed: Shira Elqayam

Reviewer: 2

Comments to the Author(s)

The paper “Does Competitive Winning Increase Subsequent Cheating?” aims at replicating an experiment conducted by Schurr & Ritov (2016) and substantially extending and testing other plausible psychological mechanisms underlying the effect.

Before commenting on the experimental design of the replication and the subsequent experiment, it is important to put things in the right context. Schurr & Ritov (2016) examine how social comparisons winners and losers make affects their sense of entitlement and subsequent asocial behaviors. In contrast to previous research, Schurr & Ritov examine what happens after the competition ends, competitors adapt to their status as winners or losers, and they go on to interact with new partners.

Schurr & Ritov claim that a key factor influencing winners’ and losers’ post-competition dishonesty is the perceived relevance of the competition to the individual’s self-image. To the extent that winning is perceived to be determined by the individual’s ability, effort, or rights, the winner, more than the loser, may feel entitled to behave in self-serving ways. In that case, winning (as opposed to losing) is likely to engender asocial behaviors such as avoiding the opportunity to help others, or dishonestly taking more than one is entitled to. However, if winning is not fairness-based, such as when it is randomly determined, the loser, rather than the winner, may act to “correct” the unfair outcome and cheat to compensate for the perceived unfair loss.

The authors propose to examine two psychological mechanisms – self-confidence and feeling lucky. Self-confidence is a reasonable mechanism that merits examination. As per feeling-lucky, there was no indication for this mechanism. If anything, Study 3a in Schurr & Ritov demonstrates that feeling lucky enhances egalitarianism among winners and does not affect losers.

The text includes several inaccuracies and wrong interpretations:

(1) Competing hypothesis - The authors propose a competing hypothesis that losing rather than winning increases subsequent cheating. To back their claim the authors refer to Zitek’s study (see page 2 lines 5-9). Note however that Zitek and her colleagues examined situations where the competition was stopped in the middle in an unfair way (Experiment 3). In this case, as noted above, the loser rather than the winner may act to level the field. Such a behavior indicates that the competitors did not adapt to their status.

(2) It is also inaccurate to frame Schurr & Ritov (2016) Study 3b as reflecting “the achievement of a purely personal goal”. The idea behind this study is to attenuate social comparisons by focusing participants on meeting a fixed threshold.

(3) The authors claim that in the trivia study participants could infer their actual performance in the study. It is unlikely the case, each question had 4 possible answers and participants did not receive a feedback after answering the questions. The questions were tricky in the sense that two answers seemed plausible. Importantly, to the extent to competition prompts relative judgment, the study was not used to test competition, but rather to test what happens when relative judgment attenuates and participants meet a fixed threshold.

(4) As per the anomalies and problems you note (page 3 line 12), increasing sample size is always adequate. Importantly, back in 2015, when the study was conducted this was the common and adequate sample size.

The replication study:

The proposed ‘replication study’ does not actually replicate the original study, due to two major flaws:

(1) Schurr & Ritov (2016) let half the participants play the dice under cup game. In the proposed replication study all the participants play the dice game. This is a deviation from the original procedure and its consequences cannot be predicted.

(2) Pairing participants with collaborators who purposefully achieve 0 or perfect score is bound to raise suspicions among the participants.

Importantly, when the original study was conducted, the experimenter flipped a coin and assigned all participants on the right or left side of the laboratory the role of dictators or recipients respectively.

The original payment was set to 12 shekels because this was the price of a cup coffee and a croissant. Do 3 pounds have the same buying power?

Finally, we are in the middle of a COVID-19 crisis. People are required to wear facial masks and practice social distancing. These requirements are likely to affect dishonesty.

The online study:

While there is indeed place to explore different processes that may account for the effect, there are three issues the authors should pay attention to:

Cheating in online studies is problematic. It is difficult to achieve liable and replicable dishonesty in online studies that were conducted on online panels. Participants do not believe that they are playing against a real person, nor do they believe that the experimenter respects and maintains complete privacy.

In addition, paying participants with vouchers increases the chances that losers will try to compensate for their lost income. Therefore, it is important to give participants a present, without telling its monetary value.

Finally, pairing participants with collaborators who purposefully achieve 0 or perfect score raises suspicions among participants.

Author's Response to Decision Letter for (RSOS-202197.R0)

See Appendix A.

RSOS-202197.R1 (Revision)

Review form: Reviewer 1 (Shira Elqayam)

Do you have any ethical concerns with this paper?

No

Recommendation?

Accept in principle

Comments to the Author(s)

The authors have done a thorough job responding to comments, revising the manuscript and strengthening what was a strong proposal to start with. I have no further comments and I am happy to recommend acceptance for publication.

Review form: Reviewer 2

Do you have any ethical concerns with this paper?

No

Recommendation?

Major revision

Comments to the Author(s)

See attached file (Appendix B).

Decision letter (RSOS-202197.R1)

Dear Dr Mangiarulo,

On behalf of the Editors, I am pleased to inform you that your Manuscript RSOS-202197.R1 entitled "Does Competitive Winning Increase Subsequent Cheating?" has been accepted in principle for publication in Royal Society Open Science subject to minor revision in accordance with the referee and editor suggestions. Please find their comments at the end of this email.

The reviewers and handling editors have recommended publication, but also suggest some minor revisions to your manuscript. Therefore, I invite you to respond to the comments and revise your manuscript.

Please you submit the revised version of your manuscript within 7 days (i.e. by the 03-Apr-2021). If you do not think you will be able to meet this date please let me know immediately.

When submitting your revised manuscript, you will be able to respond to the comments made by the referees and you should upload a file "Response to Referees". You can use this to document any changes you make to the original manuscript. In order to expedite the processing of the revised manuscript, please be as specific as possible in your response to the referees.

Full author guidelines can be found here <https://royalsocietypublishing.org/rsos/registered-reports>.

on behalf of Professor Chris Chambers (Subject Editor, Royal Society Open Science)
openscience@royalsociety.org

Associate Editor Comments to Author (Professor Chris Chambers):

Associate Editor: 1

Comments to the Author:

The two original reviewers have evaluated the revised manuscript. Reviewer 1 is now satisfied and recommends Stage 1 in-principle acceptance (IPA) whereas Reviewer 2 notes some remaining concerns to address, including a potential design flaw, factual errors and deviations from the original study. With the article now transferred to the RR article type, the scope for deviations is now wider.

Provided the authors can respond comprehensively to the remaining comments of Reviewer 2 in a final revision, then 1 IPA should be forthcoming without requiring further in-depth Stage 1 review.

Reviewer comments to Author:

Reviewer: 1

Comments to the Author(s)

The authors have done a thorough job responding to comments, revising the manuscript and strengthening what was a strong proposal to start with. I have no further comments and I am happy to recommend acceptance for publication.

Reviewer: 2

Comments to the Author(s)

See attached file

Author's Response to Decision Letter for (RSOS-202197.R1)

See Appendix C.

Decision letter (RSOS-202197.R2)

Dear Dr Mangiarulo

On behalf of the Editor, I am pleased to inform you that your Manuscript RSOS-202197.R2 entitled "Does Competitive Winning Increase Subsequent Cheating?" has been accepted in principle for publication in Royal Society Open Science.

You may now progress to Stage 2 and complete the study as approved. Before commencing data collection we ask that you:

1) Update the journal office as to the anticipated completion date of your study.

2) Register your approved protocol on the Open Science Framework (<https://osf.io/rr>) or other recognised repository, either publicly or privately under embargo until submission of the Stage 2 manuscript. Please note that a time-stamped, independent registration of the protocol is mandatory under journal policy, and manuscripts that do not conform to this requirement cannot be considered at Stage 2. The protocol should be registered unchanged from its current approved state, with the time-stamp preceding implementation of the approved study design. We recommend using the dedicated RR registration portal at <https://osf.io/rr>

Following completion of your study, we invite you to resubmit your paper for peer review as a Stage 2 Registered Report. Please note that your manuscript can still be rejected for publication at Stage 2 if the Editors consider any of the following conditions to be met:

- The results were unable to test the authors' proposed hypotheses by failing to meet the approved outcome-neutral criteria.
- The authors altered the Introduction, rationale, or hypotheses, as approved in the Stage 1 submission.
- The authors failed to adhere closely to the registered experimental procedures. Please note that any deviations from the approved experimental procedures must be communicated to the editor immediately for approval, and prior to the completion of data collection. Failure to do so can result in revocation of in-principle acceptance and rejection at Stage 2 (see complete guidelines for further information).
- Any post-hoc (unregistered) analyses were either unjustified, insufficiently caveated, or overly dominant in shaping the authors' conclusions.
- The authors' conclusions were not justified given the data obtained.

We encourage you to read the complete guidelines for authors concerning Stage 2 submissions at <https://royalsocietypublishing.org/rsos/registered-reports#ReviewerGuideRegRep>. Please especially note the requirements for data sharing, reporting the URL of the independently registered protocol, and that withdrawing your manuscript will result in publication of a Withdrawn Registration.

Once again, thank you for submitting your manuscript to Royal Society Open Science and we look forward to receiving your Stage 2 submission. If you have any questions at all, please do not hesitate to get in touch. We look forward to hearing from you shortly with the anticipated submission date for your stage two manuscript.

on behalf of Professor Chris Chambers (Registered Reports Editor, Royal Society Open Science)
openscience@royalsociety.org

Author's Response to Decision Letter for (RSOS-202197.R2)

See Appendix D.

RSOS-202197.R2 (Revision)

Review form: Reviewer 1 (Shira Elqayam)

Is the manuscript scientifically sound in its present form?

Yes

Are the interpretations and conclusions justified by the results?

Yes

Is the language acceptable?

Yes

Do you have any ethical concerns with this paper?

No

Have you any concerns about statistical analyses in this paper?

Yes

Recommendation?

Accept with minor revision

Comments to the Author(s)

Review of RSOS-202197, Colman, Pulford, Frosch, & Mangiarulo: Does Competitive Winning Increase Subsequent Cheating?

This is a pre-registered study of two experiments, of which the first one is a replication study of Schurr and Ritov (2016), and the second one is a follow-up study aiming at an explanation of whatever effects might be found in the first experiment. I have reviewed the Stage 1 manuscript, and it is a pleasure to see the findings. As before, the literature review, rationale, and hypotheses are clear and engaging. I strongly support publishing the article, but before that, I have a few relatively minor comments for improving the manuscript.

In a previous review I have commented on the use of trait scales in Study 2 to measure changes following an experimental manipulation. The latter is likely to have effects on states, less likely to affect traits, which are more stable. I would like to see this at least acknowledged as a limitation of the study.

I have several comments regarding the SEM analysis. (1) Can the authors please include a figure showing the path diagram of the SEM analysis? (2) It is customary to report a raft of measures for SEM. Just chi square is not enough. I would suggest including at least RMSEA. Additional measures can include chi square/df, CFI, and AIC. (3) Lastly, only one SEM model is reported. Have the authors considered alternative models?

Minor comments:

The use of DV's is initially confusing: pence in Study 1, GBPs in Study 2. Can this be made clear in the respective design sections please. Perhaps consider using a single measure (pence or pounds) throughout.

Section 2.2.3: How did participants collect the earbuds? This was an online study.

Section 2.2.4, end of p. 9: "a weak relationship with Inequality Aversion, $r(275) = .14$, $p = .02$, when all four conditions were combined". Given the number of analyses, this result is only significant in one-tail (Bonferroni correction $.05/3 = .0166$). Is one-tail justified in this context?

Section 2.2.5. Maybe move to an appendix.

Section 3 (Discussion), p. 12: "A third possibility is that the discrepancy between Schurr and Ritov's findings and ours arises from a cross-cultural difference between students in Israel and the UK; but we are unaware of any evidence that might support that interpretation, and if correct it would also severely limit the generality of the basic finding." I have a couple of comments here. First, this is relatively unlikely given that both the UK and Israel are WEIRD culture countries. Second, if correct, this would not just limit the generality of the original finding, it would limit the generality of the current finding in exactly the same way.

In conclusion, the authors have done excellent job running this replication and reporting the results. My recommendation is to publish the manuscript conditional on a few minor revisions.

Signed: Shira Elqayam

Review form: Reviewer 2

Is the manuscript scientifically sound in its present form?

No

Are the interpretations and conclusions justified by the results?

No

Is the language acceptable?

Yes

Do you have any ethical concerns with this paper?

No

Have you any concerns about statistical analyses in this paper?

No

Recommendation?

Major revision

Comments to the Author(s)

The authors dismissed my former concerns and evade an open matter-of-fact discussion of my main critiques. My concerns grow stronger in light of the fact that across experiments and experimental conditions (including the control) all the participants stole money. This finding suggests that the experimental manipulation was too strong or not sensitive enough to capture any differences between the experimental conditions.

My main critique is that the experimental settings differ substantially from the original study. In the new study all the participants did the dice under cup task whereas in the original study only half did the task. From a social perspective, letting half do the task while the other half's payment depends on those who do the task is a very strong manipulation. It is also likely to trigger different behaviors and tap on different psychological mechanisms. For example, letting everybody (as opposed to only half) do the task may signal to participants that a new competition

is about to begin and that everybody had better “level the field” if they want to leave the experiment with some money. This is a critical difference that questions the extent to which the replication really replicates the original study. This critique cannot be casually mentioned for the first time in the discussion and readily dismissed as “unusual circumstances” (see p.13). There are many “dictatorial” and “semi-dictatorial” real-life settings where the decisions of few (e.g., managers) affect the lives of many.

Experiment 2 also fails to replicate the original setting. Participants did not steal money from their counterparts but rather from the experimenter. Furthermore, because it is very easy to infer from the cheating task whether participants cheated or not, the Experiment seems to examine the extent to which the status of winning vs. losing impacts overt cases of dishonesty. These in turn were not hypothesized nor reported or discussed in both papers.

Notably, given that Experiment 2 tests the role of status and given that both winners and losers stole money from the experimenter, the findings also fail to replicate other research findings such as those of John, Loewenstein & Rick (2014); Zitek et al (2010) and Siniver & Yaniv 2018 – In these studies high status participants/winners did not steal less than low status participants/losers.

Decision letter (RSOS-202197.R3)

Dear Dr Mangiarulo,

On behalf of the Editor, I am pleased to inform you that your Stage 2 Registered Report RSOS-202197.R3 entitled "Does Competitive Winning Increase Subsequent Cheating?" has been deemed suitable for publication in Royal Society Open Science subject to minor revision in accordance with the referee suggestions. Please find the referees' comments at the end of this email.

The reviewers and Subject Editor have recommended publication, but also suggest some minor revisions to your manuscript. We invite you to respond to the comments and revise your manuscript. Below the referees' and Editors' comments (where applicable) we provide additional requirements. Final acceptance of your manuscript is dependent on these requirements being met. We provide guidance below to help you prepare your revision.

Please submit your revised manuscript and required files (see below) no later than 7 days from today's (ie 01-Jul-2022) date. Note: the ScholarOne system will 'lock' if submission of the revision is attempted 7 or more days after the deadline. If you do not think you will be able to meet this deadline please contact the editorial office immediately.

on behalf of Professor Chris Chambers
 (Registered Reports Editor, Royal Society Open Science)
 openscience@royalsociety.org

Associate Editor Comments to Author (Professor Chris Chambers):

The two reviewers who assessed the manuscript at Stage 1 kindly returned to evaluate the Stage 2 manuscript. As you will, Reviewer 1 is largely satisfied, noting mostly minor points for clarification and limitations for further consideration in the Discussion. Reviewer 2 is more critical, noting concerns with the design and interpretation.

As this is a Stage 2 RR, the design is not relitigated as part of Stage 2 review; therefore concerns raised at this stage cannot result in rejection. In addition, since the article was transferred during Stage 1 evaluation from the Replications article type to the Registered Reports track, the permissible scope for deviations is greater. That said, I do not wish to dismiss the concerns of this reviewer, which you should address thoroughly in your response to reviewers and in the Discussion section of the Stage 2 manuscript.

One final point: in revising, you may occasionally feel pressure to alter parts of the Introduction and Method section of the Stage 2 manuscript that were approved at Stage 1. Please make such changes *only* where doing so is necessary to correct a factual error or avoid a significant misunderstanding. In addition, please do not move material from this previously approved part of the manuscript to supplementary information (irrespective of reviewer requests).

Reviewer Comments to Author:

Reviewer: 1

Comments to the Author(s)

Review of RSOS-202197, Colman, Pulford, Frosch, & Mangiarulo: Does Competitive Winning Increase Subsequent Cheating?

This is a pre-registered study of two experiments, of which the first one is a replication study of Schurr and Ritov (2016), and the second one is a follow-up study aiming at an explanation of whatever effects might be found in the first experiment. I have reviewed the Stage 1 manuscript, and it is a pleasure to see the findings. As before, the literature review, rationale, and hypotheses are clear and engaging. I strongly support publishing the article, but before that, I have a few relatively minor comments for improving the manuscript.

In a previous review I have commented on the use of trait scales in Study 2 to measure changes following an experimental manipulation. The latter is likely to have effects on states, less likely to affect traits, which are more stable. I would like to see this at least acknowledged as a limitation of the study.

I have several comments regarding the SEM analysis. (1) Can the authors please include a figure showing the path diagram of the SEM analysis? (2) It is customary to report a raft of measures for SEM. Just chi square is not enough. I would suggest including at least RMSEA. Additional measures can include chi square/df, CFI, and AIC. (3) Lastly, only one SEM model is reported. Have the authors considered alternative models?

Minor comments:

The use of DV's is initially confusing: pence in Study 1, GBPs in Study 2. Can this be made clear in the respective design sections please. Perhaps consider using a single measure (pence or pounds) throughout.

Section 2.2.3: How did participants collect the earbuds? This was an online study.

Section 2.2.4, end of p. 9: "a weak relationship with Inequality Aversion, $r(275) = .14$, $p = .02$, when all four conditions were combined". Given the number of analyses, this result is only significant in one-tail (Bonferroni correction $.05/3=.0166$). Is one-tail justified in this context?

Section 2.2.5. Maybe move to an appendix.

Section 3 (Discussion), p. 12: "A third possibility is that the discrepancy between Schurr and Ritov's findings and ours arises from a cross-cultural difference between students in Israel and the UK; but we are unaware of any evidence that might support that interpretation, and if correct it would also severely limit the generality of the basic finding." I have a couple of comments here. First, this is relatively unlikely given that both the UK and Israel are WEIRD culture countries. Second, if correct, this would not just limit the generality of the original finding, it would limit the generality of the current finding in exactly the same way.

In conclusion, the authors have done excellent job running this replication and reporting the results. My recommendation is to publish the manuscript conditional on a few minor revisions.

Signed: Shira Elqayam

Reviewer: 2

Comments to the Author(s)

The authors dismissed my former concerns and evade an open matter-of-fact discussion of my main critiques. My concerns grow stronger in light of the fact that across experiments and experimental conditions (including the control) all the participants stole money. This finding suggests that the experimental manipulation was too strong or not sensitive enough to capture any differences between the experimental conditions.

My main critique is that the experimental settings differ substantially from the original study. In the new study all the participants did the dice under cup task whereas in the original study only half did the task. From a social perspective, letting half do the task while the other half's payment depends on those who do the task is a very strong manipulation. It is also likely to trigger different behaviors and tap on different psychological mechanisms. For example, letting everybody (as opposed to only half) do the task may signal to participants that a new competition is about to begin and that everybody had better "level the field" if they want to leave the experiment with some money. This is a critical difference that questions the extent to which the replication really replicates the original study. This critique cannot be casually mentioned for the first time in the discussion and readily dismissed as "unusual circumstances" (see p.13). There are many "dictatorial" and "semi-dictatorial" real-life settings where the decisions of few (e.g., managers) affect the lives of many.

Experiment 2 also fails to replicate the original setting. Participants did not steal money from their counterparts but rather from the experimenter. Furthermore, because it is very easy to infer from the cheating task whether participants cheated or not, the Experiment seems to examine the extent to which the status of winning vs. losing impacts overt cases of dishonesty. These in turn were not hypothesized nor reported or discussed in both papers.

Notably, given that Experiment 2 tests the role of status and given that both winners and losers stole money from the experimenter, the findings also fail to replicate other research findings such as those of John, Loewenstein & Rick (2014); Zitek et al (2010) and Siniver & Yaniv 2018 - In these studies high status participants/winners did not steal less than low status participants/losers.

===PREPARING YOUR MANUSCRIPT===

one version should clearly identify all the changes that have been made (for instance, in coloured highlight, in bold text, or tracked changes);

===PREPARING YOUR REVISION IN SCHOLARONE===

-- If you are requesting an article processing charge waiver, you must select the relevant waiver option (if requesting a discretionary waiver, the form should have been uploaded, see 'File upload' above).

-- If you have uploaded any electronic supplementary (ESM) files, please ensure you follow the guidance at <https://royalsociety.org/journals/authors/author-guidelines/#supplementary-material> to include a suitable title and informative caption. An example of appropriate titling and captioning may be found at https://figshare.com/articles/Table_S2_from_Is_there_a_trade-off_between_peak_performance_and_performance_breadth_across_temperatures_for_aerobic_scope_in_teleost_fishes_/3843624.

Author's Response to Decision Letter for (RSOS-202197.R3)

See Appendix E.

Decision letter (RSOS-202197.R4)

Dear Dr Mangiarulo:

I am pleased to inform you that your manuscript entitled "Does Competitive Winning Increase Subsequent Cheating?" is now accepted for publication in Royal Society Open Science.

Please remember to make any data sets or code libraries 'live' prior to publication, and update any links as needed when you receive a proof to check - for instance, from a private 'for review' URL to a publicly accessible 'for publication' URL. It is also good practice to add data sets, code and other digital materials to your reference list.

Royal Society Open Science is a fully open access journal. A payment may be due before your article is published. Our partner Copyright Clearance Center's RightsLink for Scientific Communications will contact the corresponding author about your open access options from the email domain @copyright.com (if you have any queries regarding fees, please see <https://royalsocietypublishing.org/rsos/charges> or contact authorfees@royalsociety.org).

on behalf of Professor Chris Chambers (Subject Editor).

Follow Royal Society Publishing on Twitter: @RSocPublishing
Follow Royal Society Publishing on Facebook:
<https://www.facebook.com/RoyalSocietyPublishing/>
Read Royal Society Publishing's blog:
<https://royalsociety.org/blog/blogsearchpage/?category=Publishing>

Appendix A

Response to Referees

In the numbered paragraphs below are responses to all the associate editor's and referees' comments, in the order of appearance in the decision letter sent by email on 18 January 2021.

Associate Editor

1. "First, as a point of editorial guidance, the authors should note that the Replication format is expressly intended for close (as possible) replications of previous studies, not extensions. Therefore, if the authors want to continue down the Replications track then Study 2 should be dropped from the proposal, and the methods of Study 1 must be aligned as closely as possible to the original methods (retaining only deviations that are unavoidable). Where there are potential flaws in the original target study, the authors should still conduct a close replication (regardless of flaws) but could then propose a tight follow-up replication study that addresses them (and is therefore slightly less close to the original study). The authors may therefore consider dropping Study 2 and splitting Study 1 into two studies -- one that is as close as possible to the target study (addressing the referees' concerns in this area), and another with minor modifications.

"I appreciate that the authors may not prefer this route. If so, and they wish to preserve the current sequence of studies, then the article should be transferred from the Replications article type to the Registered Reports article type, which provides more latitude for deviations from the original replication study while also requiring more strict criteria to be met. If the authors prefer this option then please contact the journal office to discuss next steps."

Response: We thought that our replication in Study 1 was aligned as close as possible to the original methods, but we note that Referee 2 questioned this, so we are happy to transfer the manuscript from the Replications article type to the Registered Reports article type. We have been in touch with the journal office, as suggested by the Associate Editor. Anita Kristiansen, Editorial Coordinator, advised us on 21 January 2021 that when we submit our revision, we should be able to change the Article Type to be a Registered Report, retaining all existing information from the original submission and changing only the Article Type. This is how we are proceeding.

Referee 1

2. "I think it has much to recommend it. Nevertheless, I would recommend some changes before it is ready for publication in RSOC."

Response: We are grateful for this generally positive response.

3. "The authors make a strong argument about the flaws in the original S&R study, and propose a sensible way forward for amending them. I only have a couple of very minor comments here. The first one is to query the method of obtaining exactly 252 participants, given attrition. How will the authors treat attrition and how will dropout participants be replaced and when?"

Response: This is a good point that we pondered hard when preparing the original submission. It would indeed be difficult to be sure of getting precisely 252 participants without having to drop some volunteers arbitrarily to get to that exact number. We really

wanted to write “at least 252” but were worried that, in the final article after completing the study, we would need to provide an exact figure, and at that stage we could hardly write “We recruited at least 252 undergraduate students”. However, we have now changed the manuscript to “We recruited at least 252 undergraduate students” in the hope and expectation that we will be permitted to substitute the exact figure and more natural language in the final version.

4. “My second comment is about the exchange rate between NIS and GBP – this seems to be calculated by the current (2020) rate, but the exchange rate changed dramatically post the 2016 Brexit vote, and S&R’s study might well have been conducted prior to this. With the small sums involved this might not make a huge difference, but it’s worth checking and maybe commenting on.”

Response: Referee 2 has helpfully revealed (see paragraph 15 below) that “the original payment was set to 12 shekels because this was the price of a cup coffee and a croissant”. This provides us with a convenient way of calibrating the payment to correspond as closely as possible with the original Israeli study, and the £3.00 that we had suggested, turns out to be just right. We have added the following comment to “2.1.1 Participants”: “The remuneration corresponds to the approximate price of a cup coffee and a croissant, as in the original study.”

5. “The main issues I see are with Study 2. The authors rightly criticise S&R for not providing evidence to support a causal mechanism for their explanation of the findings. However, their own proposed method also falls short of a fully causal explanation. Their hypotheses are stated clearly, and only aim at establishing association rather than causal mechanisms, but given the criticism of the previous work this is a bit disappointing. Admittedly, the proposed SEM analysis somewhat compensates for this, since SEM can establish causality, but the authors are right to be wary of claiming causality without experimental manipulation. Perhaps the best way to treat Study 2 is as a proof-of-concept study, which will need to be followed up with experimental design.”

Response: We share Referee 2’s wariness about causal interpretations of SEM, and we are grateful for the helpful suggestion. The eventual goal must surely be to provide evidence for a causal mechanism, and SEM purports to provide evidence of causality but, unlike many of our colleagues, we are reluctant to accept what is ultimately correlational data as evidence of causality. In the third-to-last paragraph before the heading “2. Materials and Methods”, we have therefore added the following: “In either case, we will examine the experimental hypotheses using techniques of structural equation modelling that are designed to test causal hypotheses but are based on purely correlational data. We offer Study 2 as a proof-of-concept investigation capable of drawing tentative conclusions, pending controlled experiments that can corroborate its findings decisively.”

6. “Another tweak I would recommend is running SEM-specific power analysis – a sample size of 252 seems a bit on the short side for SEM, especially given the dichotomous IV and the potential need for multi-group analysis.”

Response: We have reviewed the sources shown at the bottom of this response, and we also found the most widely recommended used SEM-specific power analysis calculator:

Soper, D.S. (2018). *A-priori Sample Size Calculator for Structural Equation Models*.
<https://www.danielsoper.com/statcalc/calculator.aspx?id=89>

We are grateful for this tip. It turns out, surprisingly, that the calculator's recommended sample size for a large effect is almost exactly the same as we suggested and, more surprisingly, for a small effect size it is less than our initial suggestion. This appears to be because we plan to test a very simple path model without any latent variables. The calculator does not allow for zero latent variables, so we conservatively accepted the results for one. Also, Referee 1 comments that our IV is dichotomous, but at least the group sizes are even, and this may further mitigate the required sample size somewhat. In the light of this more thorough power analysis, we have changed the sample sizes in the first paragraph of "2.2.4. Sampling and analysis plan" to "at least 100" and "at least 200".

Tabachnick, B., & Fidell, L. (2013). *Using multivariate statistics*. Boston: Pearson Education.

Westland, J. C. (2010). Lower bounds on sample size in structural equation modelling. *Electronic Commerce Research Applications*, 9(6), 476–487.

Referee 2

7. "Schurr & Ritov claim that a key factor influencing winners' and losers' post-competition dishonesty is the perceived relevance of the competition to the individual's self-image. To the extent that winning is perceived to be determined by the individual's ability, effort, or rights, the winner, more than the loser, may feel entitled to behave in self-serving ways. In that case, winning (as opposed to losing) is likely to engender asocial behaviors such as avoiding the opportunity to help others, or dishonestly taking more than one is entitled to. However, if winning is not fairness-based, such as when it is randomly determined, the loser, rather than the winner, may act to "correct" the unfair outcome and cheat to compensate for the perceived unfair loss."

Response: We understand perfectly that this was Schurr & Ritov's interpretation of their experiment, and they may be right, but they did not test this explanation empirically. One of the main objectives of our own study is to test the claimed effect rigorously in Study 1 and, if we replicate the original finding, to test Schurr & Ritov's explanation for it empirically against competing explanations in Study 2.

8. "The authors propose to examine two psychological mechanisms – self-confidence and feeling lucky. Self-confidence is a reasonable mechanism that merits examination. As per feeling-lucky, there was no indication for this mechanism. If anything, Study 3a in Schurr & Ritov demonstrates that feeling lucky enhances egalitarianism among winners and does not affect losers."

Response: Schurr and Ritov did not test their preferred mechanism (sense of entitlement) empirically. If we replicate their effect in Study 1, then we propose to test their proposed mechanisms and both of our suggested alternative mechanisms in Study 2. If Referee 2 is right and self-confidence is a reasonable explanation, then the results will confirm this. If, on the other hand, feeling lucky or sense of entitlement explains the effect better, then our Study 2 will provide empirical evidence for this.

9. "The text includes several inaccuracies and wrong interpretations:

(1) Competing hypothesis - The authors propose a competing hypothesis that losing rather than winning increases subsequent cheating. To back their claim the authors refer to Zitek's study (see page 2 lines 5-9). Note however that Zitek and her colleagues examined situations

where the competition was stopped in the middle in an unfair way (Experiment 3). In this case, as noted above, the loser rather than the winner may act to level the field. Such a behavior indicates that the competitors did not adapt to their status.”

Response: We are highly motivated to provide an article that is as accurate as possible and therefore to eliminate any inaccuracy or wrong interpretation that the manuscript may contain, but we cannot see how this can fall into either category. We state explicitly in the third paragraph of the Introduction that “Zitek et al. [9] suggested that it is a feeling of being wronged, rather than competitive winning, that tends to engender a sense of ‘victim entitlement’ ”. We cannot see any inaccuracy or wrong interpretation in this: it is exactly what Zitek et al. asserted. It is true that their participants exhibited a sense of entitlement only when their initial loss was unfair, but it is nonetheless true that they claimed that it was being wronged and not competitive winning that engenders a sense of entitlement. They wrote: “We propose instead that feeling wronged gives people a sense of entitlement to obtain positive outcomes—and to avoid negative ones—that frees them from the usual requirements of social life” (p. 245). We cannot see what here could be changed to increase accuracy.

10. “(2) It is also inaccurate to frame Schurr & Ritov (2016) Study 3b as reflecting “the achievement of a purely personal goal”. The idea behind this study is to attenuate social comparisons by focusing participants on meeting a fixed threshold.”

Response: This comment is quite right: Schurr and Ritov’s Study 3b was headed “The Effect of Achieving a Goal”, not a purely personal goal. We have deleted “purely personal” from the first paragraph of “1. Introduction”. We are grateful for this correction, because we are trying hard to ensure that our manuscript is accurate in every detail.

11. “(3) The authors claim that in the trivia study participants could infer their actual performance in the study. It is unlikely the case, each question had 4 possible answers and participants did not receive a feedback after answering the questions. The questions were tricky in the sense that two answers seemed plausible. Importantly, to the extent to competition prompts relative judgment, the study was not used to test competition, but rather to test what happens when relative judgment attenuates and participants meet a fixed threshold.”

Response: We respectfully disagree. If the questions were difficult, as stated by Schurr and Ritov, and tricky, as stated by Referee 2, then surely participants who did not know any right answers, or knew very few, must have inferred that they failed to answer more than half correctly. That raises the possibility, at least, that some of those who were randomly assigned to the successful group (more than half their answers correct) might have suspected that they were not real achievers, as we suggested. This is obviously a matter of opinion, and we are not persuaded to change ours by Referee 2’s comment. In any event, it is merely a comment in our introduction and has no bearing on the validity of our experiment. Nevertheless, in the seventh paragraph of the Introduction, we have replaced “at least some of those who were randomly assigned to the successful group must surely have been aware that they were not real achievers” with “at least some of those who were randomly assigned to the successful group may surely have been aware or suspected that they were not real achievers”.

12. “(4) As per the anomalies and problems you note (page 3 line 12), increasing sample size is always adequate. Importantly, back in 2015, when the study was conducted this was the common and adequate sample size.”

Response: Yes, we agree that, back in 2015, underpowered experiments in psychology were quite common. That is usually cited as one of the main reasons why the replication crisis hit psychology so hard.

13. “The proposed ‘replication study’ does not actually replicate the original study, due to two major flaws:

(1) Schurr & Ritov (2016) let half the participants play the dice under cup game. In the proposed replication study all the participants play the dice game. This is a deviation from the original procedure and its consequences cannot be predicted.”

Response: This is Referee 2’s most serious criticism, and it needs careful examination. From the point of view of the participants whose behaviour is investigated (the true participants), this is a distinction without a difference, and it is important to understand why it cannot have any bearing on their cheating behaviour.

- In our experiment, each participant is paired with another whose behaviour is not investigated (one of two trained participants serving as dummy co-players for all the true participants), and only the scores achieved by the true participants are recorded. But the true participants are simply told that they are each paired with another participant and do not know and have no reason to guess that their co-players are dummy participants. It follows that this fact cannot have any bearing on their cheating behaviour.
- In the S&R experiment, each participant is paired with another whose behaviour is not investigated (a “passive recipient”), and only the scores achieved by the true participants are recorded. But the true participants are simply told that they are each paired with another participant and do not know and have no reason to guess that their opponents are all passive recipients who do not play the dice game, and therefore that only half the participants play the dice game. It follows that this fact cannot have any bearing on their cheating behaviour.

It is therefore not correct to suggest that our elimination of the “passive recipients” in the dice game and their replacement with two trained dummy co-players, was a “major flaw” or indeed a flaw of any kind. It seems more correct to view it as an improvement in experimental design. In any case, because the difference in procedure involves matters that are invisible to the participants, it cannot explain any difference in results between our experiment and the original S&R experiment. We have added the following paragraph to the end of Section 2.1.4:

“It is important to point out that, from the point of view of the participants, our modification of the procedure is a distinction without a difference. Schurr and Ritov’s participants did not know that only half the participants played the dice game and were true participants, and our participants did not know that each of them was paired with one of just two trained dummy participants for the dice game. This difference relates to matters invisible to the participants and cannot explain any difference in results between Schurr and Ritov’s experiment and ours.”

14. “(2) Pairing participants with collaborators who purposefully achieve 0 or perfect score is bound to raise suspicions among the participants.

Importantly, when the original study was conducted, the experimenter flipped a coin and

assigned all participants on the right or left side of the laboratory the role of dictators or recipients respectively.”

Response: This cannot possibly raise suspicions among participants, because they will not know that their opponents achieved 0 or perfect scores. All they will know is that they won or lost against their opponents (our collaborators). In Schurr and Ritov’s original article, they wrote: “Importantly, because the estimation task is done in private, winners and losers were determined by chance to avoid selection bias” (p. 1758). Presumably, the participants did not know that their competitive success or failure was determined by chance. In our experiment, it will also be determined by chance – by whether they are randomly assigned to be paired with collaborators who deliberately let them win (by scoring zero themselves) or make them lose (by achieving a perfect score themselves) but, like Schurr and Ritov’s participants, they will not know this. From the point of view of the participants, the experience will be identical to the experience of participants in the original experiment, and that is all that matters for the validity of the replication.

15. “The original payment was set to 12 shekels because this was the price of a cup coffee and a croissant. Do 3 pounds have the same buying power?”

Response: We are grateful for this information, which confirms our choice of £3. This is roughly the price of a cup coffee and a croissant in the UK. We have added a comment to this effect in Section 2.1.1. Participants.

16. “Finally, we are in the middle of a COVID-19 crisis. People are required to wear facial masks and practice social distancing. These requirements are likely to affect dishonesty.”

Response: If participants were paired with known co-players, then the visible emotions of the other pair member could obviously influence cheating behaviour – in fact, this would be almost bound to happen. But our participants are to be paired anonymously, as in the Schurr and Ritov experiment, so this problem falls away. In any event, we plan to test participants in our Judgment and Decision Making lab, which (like most behavioural science labs) has partitions preventing participants from even seeing one another while they are performing the tasks. But this objection is largely beside the point because we are unable to run the experiment during the COVID-19 pandemic, while mask-wearing is mandatory. Our laboratory is shut and out of bounds, we are working from home, and our students are being taught remotely. This experiment will have to be run after the pandemic subsides sufficiently to allow the universities to reopen.

17. “The online study:

While there is indeed place to explore different processes that may account for the effect, there are three issues the authors should pay attention to:

Cheating in online studies is problematic. It is difficult to achieve liable and replicable dishonesty in online studies that were conducted on online panels. Participants do not believe that they are playing against a real person, nor do they believe that the experimenter respects and maintains complete privacy.”

Response: We are grateful for Referee 2’s support for this part of our proposed research. We discussed the problem of laboratory versus online cheating in the second paragraph above the heading “2. Materials and Methods”. There is an extensive body of literature on this issue, including many empirical studies, and we have thoroughly examined them. The evidence

paints a slightly more positive picture than Referee 2 appears to believe. The bottom line is this: Any differences between laboratory and online cheating will apply equally to participants on all three treatment conditions, and it is only the differences in behaviour across treatment conditions that matter; hence, as we wrote in the manuscript, “aggregate results can still reveal differences that provide valid tests of the hypotheses”.

18. “In addition, paying participants with vouchers increases the chances that losers will try to compensate for their lost income. Therefore, it is important to give participants a present, without telling its monetary value.”

Response: For the laboratory study, which is designed to replicate the original Schurr and Ritov study, we are giving participants exactly the same rewards (earbuds) as Schurr and Ritov gave their participants, and we are not telling the participants its value. This is explained in the paragraph immediately above the heading “2.1.3. Materials”. For the online study, we have to use gift vouchers, because it is impossible to send objects to online participants whose addresses are confidential, and the reward needs to be immediate. Referee 2’s fears have been addressed in the publications we cite in “2.2.1. Participants”, and they confirm that the random lottery incentive system elicits true preferences from participants.

19. “Finally, pairing participants with collaborators who purposefully achieve 0 or perfect score raises suspicions among participants.”

Response: This is the same as point 14 above, where we replied as follows: This cannot possibly raise suspicions among participants, because they will not know that their opponents achieved 0 or perfect scores. All they will know is that they won or lost against their opponents (our collaborators). In Schurr and Ritov’s original article, they wrote: “Importantly, because the estimation task is done in private, winners and losers were determined by chance to avoid selection bias” (p. 1758). Presumably, the participants did not know that their competitive success or failure was determined by chance. In our experiment, it will also be determined by chance – by whether they are randomly assigned to be paired with collaborators who deliberately let them win (by scoring zero themselves) or make them lose (by achieving a perfect score themselves) but, like Schurr and Ritov’s participants, they will not know this. From the point of view of the participants, the experience will be identical to the experience of participants in the original experiment, and that is all that matters for the validity of the replication.

Our own edits

In addition to the changes detailed above, we have made a small number of cosmetic improvements to the text and have added a footnote stating that *PNAS* would not accept a registered replication.

Appendix B

Although you did not address most of the comments I pointed out in my previous review, I'd like to still emphasize the following concerning the design of Study 1.

1. You note that "...They were instructed to report their total score from the two dice under the cup. They were told that they would be paid between 25 pence and £3.00 depending on their score (25 pence per die spot), and that the rest of the money would be paid to their dyad opponents."
 - a. In the original study participants playing the dice under cup game paid themselves from the envelop and left the lab without reporting their earnings to the experimenter. The experimenter then counted how much money was left in each envelop and distributed the envelopes to participants who did not play the game.
 - b. In the original study participants inferred the payment from the outcome of the roll. Specifically, participants were told that "You have received a cup with a small hole in the bottom, two dice, and an envelope containing 12 NIS in coins of 1 NIS each. Please turn the cup over the dice and shake it vigorously. The outcome of the shake (the numbers that came out) belongs to you. The rest of the money will go to one of the participants sitting in the lab who did not play the two-dice-under-a-cup game." It is important that you repeat the specific instructions. Mentioning specific sums may anchor/affect participants' choices.
2. You note that "Schurr and Ritov's [1] participants did not know that only half the participants played the dice game."

This is wrong! Participants at Schurr and Ritov's study knew that exactly half played the dice under cup game and half did not.

3. Do winners and losers complete the dice task in the same experimental session? In the original study winners and losers completed the dice task in the same experimental session.
4. It seems participants in the replication study participate in one long session. If this is the case, then it is an important distinction between the studies. In the original study the dice task was a distinct and different experiment.

Appendix C

Responses to Referees 2

Listed below are responses to all the associate editor's and referees' comments, following the decision letter sent by email on 26 March 2021.

Associate Editor: 1

A. "The two original reviewers have evaluated the revised manuscript. Reviewer 1 is now satisfied and recommends Stage 1 in- principle acceptance (IPA) whereas Reviewer 2 notes some remaining concerns to address, including a potential design flaw, factual errors and deviations from the original study. With the article now transferred to the RR article type, the scope for deviations is now wider."

Response: We shall address the potential design flaw and factual error in our responses to Reviewer 2's comments. We understand the above comment from the Associate Editor to imply that our replication does not have to duplicate the original experiment in every detail, including aspects of procedure that have no bearing on the validity of the replication as a test the findings reported.

B. "The authors have done a thorough job responding to comments, revising the manuscript and strengthening what was a strong proposal to start with. I have no further comments and I am happy to recommend acceptance for publication."

Response: We are grateful for this essentially supportive comment.

Reviewer: 2

1. "You note that '...They [the participants] were instructed to report their total score from the two dice under the cup. They were told that they would be paid between 25 pence and £3.00 depending on their score (25 pence per die spot), and that the rest of the money would be paid to their dyad opponents'

"a. In the original study participants playing the dice under cup game paid themselves from the envelop and left the lab without reporting their earnings to the experimenter. The experimenter then counted how much money was left in each envelop and distributed the envelops to participants who did not play the game."

Response: We are very grateful for this important clarification, because the published article seems to suggest that the participants reported their dice-roll totals in addition to taking money from their envelopes according to their claimed totals. We have therefore decided to follow the same procedure as the original, because it is likely to affect cheating: in particular, participants may be more tempted to cheat when they can do so without lying. We have added the following footnote to the end of the third paragraph of 2.1.4 to make this explicit and clear: "According to Schurr and Ritov (2016), 'First, we examined the amount of self-reported winnings participants claim in the dice-under-a-cup task' (p. 1754), and 'The participant places the cup over the dice, shakes it, and receives money according to his or her reporting of the outcome of the roll' (pp. 1754–1755). However, an anonymous reviewer with detailed knowledge of the original study has revealed that the participants did not actually

report the outcomes of the rolls other than by removing money from their envelopes: ‘In the original study participants playing the dice under cup game paid themselves from the envelop [sic] and left the lab without reporting their earnings to the experimenter. The experimenter then counted how much money was left in each envelop and distributed the envelops [sic]’ (reviewer’s comment received on 26 March, 2021). We decided to follow the same procedure, because participants may be more tempted to cheat when they can do so without lying and can justify it as carelessness rather than dishonesty.”

“b. In the original study participants inferred the payment from the outcome of the roll. Specifically, participants were told that ‘You have received a cup with a small hole in the bottom, two dice, and an envelope containing 12 NIS in coins of 1 NIS each. Please turn the cup over the dice and shake it vigorously. The outcome of the shake (the numbers that came out) belongs to you. The rest of the money will go to one of the participants sitting in the lab who did not play the two-dice-under-a-cup game.’ It is important that you repeat the specific instructions. Mentioning specific sums may anchor/affect participants’ choices.”

Response: We did in fact mention specific sums to anchor/affect participants’ responses in the participant instructions in the previous version of the manuscript. We have now made this even clearer in Section 2.1.4, third paragraph, by specifying verbatim instructions that are the same as Schurr and Ritov’s in all essential details: “You have received a cup with a small hole in its base, two dice, and an envelope containing £3.00 in 20p and 5p coins. Please invert the cup over the dice, shake it vigorously, and note the total dice spots by looking through the hole. Then take between 25 pence and £3.00 depending on your total score (25 pence per die spot) from the envelope. The rest of the money will be handed to another participant in the room.”

2. “You note that ‘Schurr and Ritov’s [1] participants did not know that only half the participants played the dice game.’ This is wrong! Participants at Schurr and Ritov’s study knew that exactly half played the dice under cup game and half did not.”

Response: We are grateful for this information, which was not clear in Schurr and Ritov’s original article. With the article now transferred to the RR article type, we understand that we are free to streamline the procedure. We have removed the incorrect assertion and have substituted, in the last paragraph of Section 2.1.4: “Our participants did not know that each of them was paired with one of just two trained dummy participants for the dice game. This difference in procedure relates to something entirely hidden from the participants that cannot explain any differences between the results of Schurr and Ritov’s experiment and ours.”

3. “Do winners and losers complete the dice task in the same experimental session? In the original study winners and losers completed the dice task in the same experimental session.”

Response: Yes, they do. We have added the following sentence to the penultimate paragraph of Section 2.1.4 to make this explicit: “Winners and losers were tested together in the same experimental sessions, as in Schurr and Ritov’s [1] original experiment, and control group participants were also tested in the same sessions.”

4. “It seems participants in the replication study participate in one long session. If this is the case, then it is an important distinction between the studies. In the original study the dice task was a distinct and different experiment.”

Response: This is indeed a small but possibly important difference. We have appended the following sentence to the first paragraph of Section 2.1.4: “All participants were told at the beginning of each testing session that they would be completing two entirely different tasks during the session. They completed Haran et al.’s perception task [23] immediately before the dice game, but the instructions for the dice game made clear that it was a distinct and different task.”

Appendix D

UNIVERSITY OF
LEICESTER

Professor Chris Chambers
Registered Reports Editor
Royal Society Open Science

29 April 2022

**Department of Neuroscience,
Psychology and Behaviour**

University Road
Leicester LE1 7RH
UK

Tel: +44 (0) 116 229 7191
Professor Andrew M. Colman
e-mail: amc@le.ac.uk

Dear Professor Chambers

Does Competitive Winning Increase Subsequent Cheating? A Registered Report Stage 2 Manuscript

Many thanks for your email dated 13 April 2021 confirming that our Manuscript RSOS-202197.R2, entitled “Does Competitive Winning Increase Subsequent Cheating?” has been accepted in principle for publication in *Royal Society Open Science*. We notified *RSOS* at the time that we expected to complete our study by the beginning of April 2022, and in the event we are slightly late, for reasons to do with lockdowns and related computer problems explained in the laboratory log that we have included in the material submitted to the OSF repository. Another reason for the slight delay is that the SPSS-based Amos software that we had planned to use for structural equation modelling turned out to be incapable of handling categorical variables. We recruited Jeremy Miles, a leading authority on structural equation modelling, as a fifth co-author after he offered to perform this analysis for us using a different R-based Lavaan software.

I wish to draw attention to two ways in which our Study 2 deviated from the approved methods and analyses of the Stage 1 submission. First, while setting up Study 2, we noticed that the control group envisaged in our Stage 1 submission had omitted a minor extraneous variable that ought to have been controlled (matched with the experimental groups) but was not. Our Study 2 experimental design therefore included the originally planned control group plus an additional control group, with the extraneous variable controlled. We did not exactly deviate from the Stage 1 proposal: we followed it faithfully but we added to it to increase rigour, and we explained this in the manuscript at the beginning of Section 2.2.

Second, the Stage 1 submission stated that the structural equation model (SEM) in Study 2 would depend on the outcome of our Study 1 experiment as follows. (a) If Study 1 confirmed Shurr and Ritov’s finding that winning leads to cheating, then the SEM would include three specified variables. (b) If, on the other hand, Study 1 found that losing leads to cheating, then the SEM would include two specified variables, one of them different from the three in (a). In the event, Study 1 found that neither winning nor losing significantly affected cheating, so we were unable to follow either plan (a) or (b). We therefore included in the SEM in Study 2 all

the measured variables that we would have included in either (a) or (b). We explained this unavoidable deviation from the original plan in the manuscript, in the final paragraph of Section 2.2.2.

I can confirm the following:

(a) Page 12 of our Stage 2 manuscript contains the URL of the OSF repository for our archived study data, digital materials/code and the laboratory log;

(b) Page 12 of our Stage 2 manuscript contains the URL of the OSF repository for the approved Stage 1 protocol;

(c) No data included at Stage 1 were subjected to the pre-registered analyses prior to the date of IPA. We had no data at that time, not even pilot data.

- The completed experiments have been executed and analysed in the manner originally approved, with only very minor unforeseen changes in those approved methods and analyses clearly noted.
- Raw and processed data have been made freely available in the OSF archive and the manuscript.
- In each of our two Results sections, we report the outcome of all registered analyses together. For Study 2, we also report pre-specified outcome-neutral quality checks and unregistered (post hoc) analyses in a brief “Section 2.2.5. Unregistered post hoc results”.
- Our Stage 2 manuscript fully reports all data analysis procedures and states that we did not eliminate data from any of the participants who performed the tasks in either of our experiments.
- We did not collect any pilot data.
- Our Stage 2 manuscript includes the specified sections: Ethical statement; Data Accessibility; Competing Interests; Authors’ Contributions; Acknowledgements; Funding; References. Figure and table captions are currently attached to the two figures and the single table in the manuscript.

I look forward to hearing from you in due course.

Yours sincerely

Andrew M. Colman

Appendix E

Responses to Reviews S2R RSOS-202197.R3

Set out below are responses to all the comments from the Associate Editor and reviewers in the decision letter sent by email on 1 July 2022.

Associate Editor

1. “As this is a Stage 2 RR, the design is not relitigated as part of Stage 2 review; therefore concerns raised at this stage cannot result in rejection. In addition, since the article was transferred during Stage 1 evaluation from the Replications article type to the Registered Reports track, the permissible scope for deviations is greater. That said, I do not wish to dismiss the concerns of this reviewer, which you should address thoroughly in your response to reviewers and in the Discussion section of the Stage 2 manuscript.”

Response: We are grateful for this clarification, and we have endeavoured to address all the reviewers’ concerns as thoroughly as possible.

2. “One final point: in revising, you may occasionally feel pressure to alter parts of the Introduction and Method section of the Stage 2 manuscript that were approved at Stage 1. Please make such changes *only* where doing so is necessary to correct a factual error or avoid a significant misunderstanding. In addition, please do not move material from this previously approved part of the manuscript to supplementary information (irrespective of reviewer requests).”

Response: We have complied rigorously with this requirement, and all alterations are shown in the version of the manuscript with tracked changes.

Reviewer 1

3. “This is a pre- registered study of two experiments, of which the first one is a replication study of Schurr and Ritov (2016), and the second one is a follow- up study aiming at an explanation of whatever effects might be found in the first experiment. I have reviewed the Stage 1 manuscript, and it is a pleasure to see the findings. As before, the literature review, rationale, and hypotheses are clear and engaging. I strongly support publishing the article, but before that, I have a few relatively minor comments for improving the manuscript.”

Response: We are grateful for these supportive comments.

4. “In a previous review I have commented on the use of trait scales in Study 2 to measure changes following an experimental manipulation. The latter is likely to have effects on states, less likely to affect traits, which are more stable. I would like to see this at least acknowledged as a limitation of the study.”

Response: Traits may help to explain cheating quite independently of winning or losing. Our thinking about this may not have been very clear until we implemented the structural equation model, and we certainly did not express the ideas clearly enough. We have added a paragraph to the end of Section 3 Discussion clarifying this. The first two sentences are: “The aim of Study 2 was to discover variables, possibly but not necessarily including competitive winning or losing, that might explain

cheating in a subsequent game of chance. The structural equation modelling should reveal whether, and if so how, winning or losing is implicated.”

5. “I have several comments regarding the SEM analysis. (1) Can the authors please include a figure showing the path diagram of the SEM analysis? (2) It is customary to report a raft of measures for SEM. Just chi square is not enough. I would suggest including at least RMSEA. Additional measures can include chi square/df, CFI, and AIC. (3) Lastly, only one SEM model is reported. Have the authors considered alternative models?”

Response: We have included a path diagram (Figure 3), as requested. We have added the RMSEA, CFI, and SRMR measures of model fit. (The AIC is 2911.24, however we do not think this is meaningful in the single model context.) We considered only one model seriously, and because it fitted well, we did not need to consider others. (We discussed the feasibility of a slightly different model to begin with, but it turned out to be unidentified and there are no results to report.)

6. “The use of DV’s is initially confusing: pence in Study 1, GBPs in Study 2. Can this be made clear in the respective design sections please. Perhaps consider using a single measure (pence or pounds) throughout.”

Response: We are also mindful of the Editor’s requirement not to make changes to text that was approved at Stage 1. We do not believe that changing the DV units of one study to match the other would make the paper clearer, and it would cause mismatches with materials and the datasets in the OSF. The DV in the first study was the money taken and it ranged from zero pence to 300 pence (£3). All of our data files refer to pence and changing the numbers from say $M = 206.27$ to $M = £0.21$ would not make it clearer. We increased the incentive payments by a factor of about 10 when we did Study 2, because the Study 1 payments were abnormally low to match Schurr & Ritov’s. We kept the full range of the scales on the Y axes of graphs 1 & 2 so that these figures would be directly comparable.

7. “Section 2.2.3: How did participants collect the earbuds? This was an online study.”

Response: In Section 2.2.3, first paragraph we wrote: “Each of the winners was informed that they had been rewarded with a pair of earbuds that they should collect after the testing session.” Although it was an online study, the participants were all students at the same university, and they were instructed to collect the earbuds from the General Office in the School of Psychology. We have not added this clarification to the manuscript, because we are not certain that it meets the threshold set by the Associate Editor in paragraph 2 above: “Please make such changes *only* where doing so is necessary to correct a factual error or avoid a significant misunderstanding.”

8. “Section 2.2.4, end of p. 9: "a weak relationship with Inequality Aversion, $r(275) = .14$, $p = .02$, when all four conditions were combined". Given the number of analyses, this result is only significant in one- tail (Bonferroni correction $.05/3 = .0166$). Is one- tail justified in this context?”

Response: Bonferroni correction was developed for use in multiple comparisons following analysis of variance, and is used to ensure that the nominal Type I error rate is at or below the value of alpha that is used. In post-hoc contrasts in analysis of variance, this is true, because the contrasts are (or very nearly are) orthogonal. However, in a study with multiple outcomes this is not the case, and so the Bonferroni-corrected alpha can be very much below alpha and unreasonably severe. Even if this were not the case, we would still be averse to using this form of correction. One of us has written about problems with using this form of correction (Miles and Banyard, 2007), and many other examples can be found in the literature. Specifically, Perneger (1998) wrote in the *British Medical Journal* that “Bonferroni adjustments are, at best, unnecessary and, at worst, deleterious to sound statistical inference” (p. 1236). The *p*-value that would emerge from a Bonferroni-adjusted test of this hypothesis would be testing against the null hypothesis for all outcomes simultaneously, but this multivariate null hypothesis is not of interest to us.

9. “Section 2.2.5. Maybe move to an appendix.”

Response: In paragraph 2 above, the Associate Editor has instructed us: “please do not move material from this previously approved part of the manuscript to supplementary information (irrespective of reviewer requests).”

10. “Section 3 (Discussion), p. 12: “A third possibility is that the discrepancy between Schurr and Ritov’s findings and ours arises from a cross- cultural difference between students in Israel and the UK; but we are unaware of any evidence that might support that interpretation, and if correct it would also severely limit the generality of the basic finding.” I have a couple of comments here. First, this is relatively unlikely given that both the UK and Israel are WEIRD culture countries. Second, if correct, this would not just limit the generality of the original finding, it would limit the generality of the current finding in exactly the same way.”

Response: These are both excellent points, and we have incorporated them at the end of the fourth paragraph of Section 3 Discussion: “A third possibility is that the discrepancy between Schurr and Ritov’s findings and ours arises from a cross-cultural difference between students in Israel and the UK; but we are unaware of any evidence that might support that interpretation, it is very unlikely given that Israel and the UK are both WEIRD cultures [38], and if correct it would severely limit the generality of Schurr and Ritov’s basic finding (and also, by symmetry, our own basic finding).”

11. “In conclusion, the authors have done excellent job running this replication and reporting the results. My recommendation is to publish the manuscript conditional on a few minor revisions.”

Response: We are grateful for Reviewer 1’s help in improving our manuscript.

Reviewer 2

12. “The authors dismissed my former concerns and evade an open matter- of- fact discussion of my main critiques.

Response: We feel that this is manifestly unfair. The suggestion that we dismissed Reviewer 2’s former concerns is, fortunately, easily disproved by glancing at our first “Response to Referees”, which confirms that we responded to every one of this

reviewers 12 concerns. Far from dismissing these concerns, we tried our best to address them, even to the point of modifying the actual experimental procedure to conform to Schurr and Ritov's procedure after learning from Reviewer 2 that the participants did not declare their dice scores to the experimenter, as implied in the published article, but merely took the money owed to them from an envelope, leaving the rest of the money behind when they left the room. We did not at any point attempt to evade an open matter-of-fact discussion of Reviewer 2's main critiques, and neither will we do so in the paragraphs that follow.

13. "My concerns grow stronger in light of the fact that across experiments and experimental conditions (including the control) all the participants stole money. This finding suggests that the experimental manipulation was too strong or not sensitive enough to capture any differences between the experimental conditions."

Response: We have never before seen an experimental manipulation being criticized for being too strong to reveal a significant difference caused by that manipulation. The implication is that the manipulation would have revealed the difference if it had been weaker. This is presumably what the reviewer means by adding that the manipulation may have been "not sensitive enough" to elicit the difference that was elicited by Schurr and Ritov. However, in both of our experiments, we manipulated the independent variable *precisely* as Schurr and Ritov did, by telling exactly half the participants, selected at random, that they had won and half that they had lost, and even using the same Haran test as the task on which they won or lost. There is no difference whatsoever in our experimental manipulation and Schurr and Ritov's. To clarify this point further, we have modified the penultimate sentence of the first paragraph of Section 3 Discussion to read: "We observed significant levels of cheating in both experimental and control conditions but failed to replicate Schurr and Ritov's basic finding of higher cheating by winners, **although the experimental manipulation of winning or losing in both of our experiments was identical to Schurr and Ritov's.**"

14. "My main critique is that the experimental settings differ substantially from the original study. In the new study all the participants did the dice under cup task whereas in the original study only half did the task. From a social perspective, letting half do the task while the other half's payment depends on those who do the task is a very strong manipulation. It is also likely to trigger different behaviors and tap on different psychological mechanisms. For example, letting everybody (as opposed to only half) do the task may signal to participants that a new competition is about to begin and that everybody had better "level the field" if they want to leave the experiment with some money. This is a critical difference that questions the extent to which the replication really replicates the original study. This critique cannot be casually mentioned for the first time in the discussion and readily dismissed as "unusual circumstances" (see p.13). There are many "dictatorial" and "semi-dictatorial" real-life settings where the decisions of few (e.g., managers) affect the lives of many."

Response: We responded at length to this criticism in our first "Response to Referees". Our participants in Study 1 and Schurr and Ritov's participants were told that the money left in the envelopes after playing the two-dice-under-a-cup game would go to another participant in the room, so from their point of view the difference in design between the experiments—the fact that we had only two passive recipients whereas Schurr and Ritov had many—was something the participants did not know. It

is not true that we mentioned this issue for the first time in the Discussion; we also devoted the third paragraph of Section 2.1.4 Procedure to it. We did indeed return to it in the Discussion, where we conceded that in our experiment the participants who cheated, apart from those in the unpaired control group treatment condition, presumably thought they were taking money from others who were fully engaged in the same activity as themselves, whereas in Schurr and Ritov's experiment they thought they were taking it from passive recipients who had done nothing for what they received, and we described the latter as "unusual circumstances". It is indeed very unusual for people who cheat to know that their victims are entirely passive and have done nothing for the money they are being cheated out of, and if that explains the difference in the findings between the two experiments, it suggests that Schurr and Ritov's finding depends on an unusual and artificial experimental setup. We have added the following sentence to the fourth paragraph of Section 3 Discussion to clarify our meaning: "In everyday life, people who cheat rarely if ever know that their victims have done nothing to earn the money out of which they are being cheated". It is worth noting that Reviewer 2's fear that this difference between Schurr and Ritov's study and ours might make a significant difference to the results turns out to be unfounded: Figure 2 shows that it made no noticeable difference at all.

We do not understand the Reviewer's argument that these circumstances are not unusual: "There are many 'dictatorial' and 'semi- dictatorial' real- life settings where the decisions of few (e.g., managers) affect the lives of many." In Schurr and Ritov's experiment, decisions of half the participants affected the lives of the other half, and in our experiment decisions of many participants affected the lives of just two, although the participants did not know that. In neither case was there anything resembling dictatorial decision making. The difference between the two experiments is not that Schurr and Ritov's cheaters were dictatorial and ours were not; it is that Schurr and Ritov's cheaters took money that should have gone to passive recipients who had done nothing to earn the money (an unusual circumstance) whereas our participants took money that they presumably thought should have gone to active participants just like themselves. Although this is Reviewer 2's "main critique", we will not pursue it any further, because it is an argument about experimental design, and the Associate Editor stipulated in Comment 1 above: "As this is a Stage 2 RR, the design is not relitigated as part of Stage 2 review."

15. "Experiment 2 also fails to replicate the original setting. Participants did not steal money from their counterparts but rather from the experimenter."

Response: We studied the behaviour of participants who stole money from their counterparts in both of our studies, and we are surprised that Reviewer 2 appears to have misunderstood the second experiment so fundamentally. In the first paragraph of Section 2.2 we stated: "Thus, in addition to our unpaired control group (Control 1) we decided to add a second paired control group (Control 2) to Study 2 with an unspecified co-player who lost money whenever a participant cheated." In the first paragraph of Section 2.2.3 we referred to "a co-player who would receive remaining money not claimed by the participant". In the second paragraph of Section 3 Discussion, we stated: "There was no significant difference in cheating between our paired and unpaired control conditions—whether cheating was associated with money being taken from another participant or from the experimenter." To eliminate any possibility of other readers understanding this as Reviewer 2 appears to have done, we

have inserted the following small addition (shown in bold here) to the beginning of the second paragraph of Section 2.2.3 Procedure: “After completing the perception task and (apart from the control groups) being designated as winners and losers, all participants were then told they were each being re-paired with a different **person to play the coin-flip task and that the rest of the money, after they took their winnings, would go to the other person that they were paired with (except in the unpaired control condition).**” We believe that this falls within the Associate Editor’s stricture in paragraph 2 above: “Please make such changes *only* where doing so is necessary to correct a factual error or avoid a significant misunderstanding.” This was a significant misunderstanding.

16. “Furthermore, because it is very easy to infer from the cheating task whether participants cheated or not, the Experiment seems to examine the extent to which the status of winning vs. losing impacts overt cases of dishonesty. These in turn were not hypothesized nor reported or discussed in both papers.”

Response: It was impossible to infer from the cheating task whether particular participants cheated or not. In Study 1, the cheating task was identical to Schurr and Ritov’s; in Study 2, it was very similar and participants were online, making it doubly impossible for the experimenter to observe overt dishonesty. In neither case was there any way of determining which participants cheated. Exactly as in Schurr and Ritov’s experiment, we could judge only statistically what proportion of participants cheated. We have not added any clarification about this to the Discussion, because we do not understand what Reviewer 2 means by saying that “it was very easy to infer cheating” and we cannot think of any way of making it clearer than it already is that individual cheating was entirely covert in both Study 1 and Study 2.

17. “Notably, given that Experiment 2 tests the role of status and given that both winners and losers stole money from the experimenter, the findings also fail to replicate other research findings such as those of John, Loewenstein & Rick (2014); Zitek et al (2010) and Siniver & Yaniv 2018 – In these studies high status participants/winners did not steal less than low status participants/losers.”

Response: Our Study 2 did not include any investigation of the role of status. The three papers cited by Referee 2 are all already discussed in our manuscript. None of them investigated the effect of competitive winning or losing in a skill-based task on subsequent cheating in a subsequent game of chance. We did not attempt to replicate any of them.